# Acidic media enables oxygen-tolerant electrosynthesis of multicarbon products from simulated flue gas

Meng Wang[1,2,5], Bingqing Wang [1,5] ✉, Jiguang Zhang[1,2,5], Shibo Xi[4], Ning Ling[1], Ziyu Mi[4], Qin Yang[1], Mingsheng Zhang[2], Wan Ru Leow [4], Jia Zhang [3] & Yanwei Lum [1,2] ✉

Renewable electricity powered electrochemical $CO_2$ reduction ($CO_2$R) offers a valuable method to close the carbon cycle and reduce our overreliance on fossil fuels. However, high purity $CO_2$ is usually required as feedstock, which potentially decreases the feasibility and economic viability of the process. Direct conversion of flue gas is an attractive option but is challenging due to the low $CO_2$ concentration and the presence of $O_2$ impurities. As a result, up to 99% of the applied current can be lost towards the undesired oxygen reduction reaction (ORR). Here, we show that acidic electrolyte can significantly suppress ORR on Cu, enabling generation of multicarbon products from simulated flue gas. Using a composite Cu and carbon supported single-atom Ni tandem electrocatalyst, we achieved a multicarbon Faradaic efficiency of 46.5% at 200 mA cm⁻², which is ~20 times higher than bare Cu under alkaline conditions. We also demonstrate stable performance for 24 h with a multicarbon product full-cell energy efficiency of 14.6%. Strikingly, this result is comparable to previously reported acidic $CO_2$R systems using pure $CO_2$. Our findings demonstrate a potential pathway towards designing efficient electrolyzers for direct conversion of flue gas to value-added chemicals and fuels.

Rapidly rising anthropogenic $CO_2$ emissions have led to serious concerns over global warming and climate change issues[1,2]. There is therefore an urgent need to develop new technologies that can efficiently capture, store and utilize $CO_2$[3]. To this end, renewable electricity powered electrochemical $CO_2$ reduction ($CO_2$R) offers a valuable method to close the carbon cycle and reduce modern society's overreliance on fossil fuels[4–12]. By transforming waste $CO_2$ into value-added chemicals and fuels, this technology can also help establish a sustainable pathway towards net-zero chemical production and store renewable energy in the form of chemical bonds[13–18]. In

particular, the production of multicarbon ($C_{2+}$) molecules such as ethylene and ethanol are attractive due to their large market size and high carbon footprint[2,19].

While the majority of literature reports use pure $CO_2$ as feedstock, it is essential to note that the cost of purifying $CO_2$ from flue gas can amount to \$70–\$100 per ton[20–22]. Depending on the product of interest, this can constitute a substantial portion of operating costs (~30% for ethylene)[9,20]. There is therefore interest in developing $CO_2$R electrolyzers that can directly convert the $CO_2$ in flue gas streams to the desired products. However, this is challenging because typical flue gas

[1]Department of Chemical and Biomolecular Engineering, National University of Singapore, Singapore 117585, Republic of Singapore. [2]Institute of Materials Research and Engineering, Agency for Science, Technology and Research (A*STAR), 2 Fusionopolis Way, Innovis #08-03, Singapore 138634, Republic of Singapore. [3]Institute of High Performance Computing, Agency for Science, Technology, and Research (A*STAR), 1 Fusionopolis Way, #16-16 Connexis, Singapore 138632, Republic of Singapore. [4]Institute of Sustainability for Chemicals, Energy and Environment (ISCE2), Agency for Science, Technology and Research (A*STAR), 1 Pesek Road, Singapore 627833, Republic of Singapore. [5]These authors contributed equally: Meng Wang, Bingqing Wang, Jiguang Zhang. ✉e-mail: wangbq@nus.edu.sg; lumyw@nus.edu.sg

contains relatively low concentrations of $CO_2$ (-15% v/v) and non-negligible amounts of $O_2$ (≥3% v/v)[20]. This inevitably results in $CO_2$ mass transport limitations, which promotes the undesired hydrogen evolution reaction (HER)[23,24]. More importantly, the presence of $O_2$ impurities can result in up to 99% of applied current being lost to the much more thermodynamically favorable oxygen reduction reaction (ORR)[20,25–27].

To address this issue, Wang and co-workers coated their cobalt phthalocyanine catalysts using a selectively permeable polymer[25] with a $CO_2/O_2$ selectivity of -20. As a result, they achieved a 75.9% Faradaic efficiency (FE) for CO production when a $CO_2$ feedstock containing 5% $O_2$ was employed. In another study, Sinton and co-workers employed ionomer coatings that selectively slowed down $O_2$ mass transport[20]. This enabled $C_{2+}$ products to be generated with a FE of 68% under high pressure (10 bar) conditions with flue gas containing $O_2$ (4% v/v) and $CO_2$ (15% v/v). Although these selective mass transport control strategies have been relatively successful, it is important to explore other avenues that could be simple yet effective for enabling direct conversion of $CO_2$ in flue gas.

In this work, we discovered that (1) choice of electrolyte and (2) catalyst design can be used to enable oxygen-tolerant production of $C_{2+}$ products in simulated flue gas. Specifically, we found that the use of acidic electrolyte results in a significant suppression of ORR activity as compared to alkaline electrolyte. This was demonstrated to be the case for both a pure Cu catalyst and a carbon-supported single-atom Ni catalyst. Density functional theory (DFT) simulations suggest that this is due to an increase in the free energy change of the rate-determining step for ORR on each of these catalysts in acidic media. We then created a composite system by coating the single-atom Ni onto Cu, creating a tandem electrocatalyst which was able to facilitate the direct conversion of $CO_2$ in simulated flue gas to $C_{2+}$ products with an FE of 46.5% at 200 mA cm$^{-2}$ in acidic electrolyte. Under these conditions, we achieved up to 24 h of stable operation with a 14.6% full-cell energy efficiency (EE) towards $C_{2+}$ products.

## Results

Our first objective in this work was to design a catalyst capable of generating $C_{2+}$ products with high selectivity in acidic media. The catalyst testing and optimization process was to be initially conducted using pure $CO_2$ feedstock, before proceeding on to our simulated flue gas experiments. Thus, we began by using magnetron sputtering to deposit 200 nm thick Cu films onto porous hydrophobic polytetrafluoroethylene (PTFE) substrates for use as $CO_2R$ gas diffusion electrodes. Representative top-down scanning electron microscopy (SEM) images (Fig. 1a and Fig. S1) show an open web-like morphology of interconnected PTFE fibers coated conformally with the Cu catalyst. These catalysts were then characterized using X-ray diffraction (XRD), where we observed Cu (111) to be the dominant facet (Fig. S2). This was found to be consistent with our Pb underpotential deposition (UPD) experiments (Fig. S3). X-ray photoelectron spectroscopy (XPS) measurements (Fig. S4) showed the expected peaks in the Cu 2$p$ narrow scans, with the presence of Cu oxides due to inevitable oxidation of the surface from exposure to ambient air[28] (Fig. S5).

$CO_2R$ was then carried out with these Cu PTFE catalysts in acidic electrolyte (0.05 M $H_2SO_4$ and 0.5 M $K_2SO_4$) at constant cathodic current densities of 100, 200, 400, and 600 mA cm$^{-2}$. These experiments were performed using a custom-made gas diffusion electrochemical flow cell system, with a similar design to that previously reported in the literature[29]. The results in Fig. S6 show that a substantial amount of hydrogen evolution reaction (HER) occurs. For instance, at 600 mA cm$^{-2}$ we observe a HER FE of 59.2% and a $C_{2+}$ FE of only 17.9% (Table S1-2). The HER FE tends to decrease with the current density and was lowest at 100 mA cm$^{-2}$ with a value of 14.3%. However, the $C_{2+}$ FE was only 53.3% which is lower than state-of-the-art acidic $CO_2R$ systems reported in the literature[30–39].

Compared to alkaline and neutral media, the $C_{2+}$ product FE tends to be lower in acidic electrolyte. This has been proposed to be due to an enhanced HER activity, less facile C–C coupling kinetics and a weakened *CO binding energy[32,36,40,41]. This can in principle, be

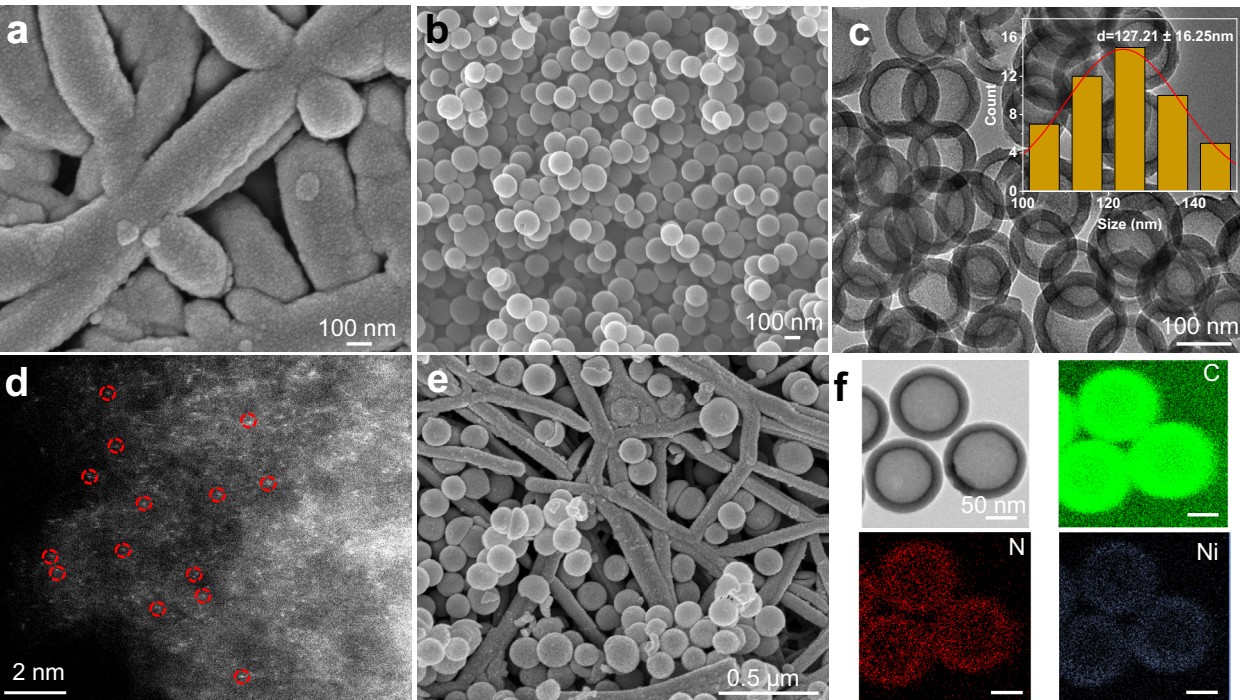

**Fig. 1 | Catalyst morphology and characterization. a** SEM images of Cu sputtered onto porous hydrophobic PTFE gas diffusion substrates. **b** SEM and (**c**) TEM images of Ni−N$_4$, consisting of Ni single-atoms hosted on a carbon support. **d** AC-HAADF-STEM image of Ni−N$_4$. **e** SEM images of the Cu PTFE/Ni−N$_4$ composite catalyst. **f** HAADF-STEM and corresponding EDS mapping images of the Ni−N$_4$ catalyst.

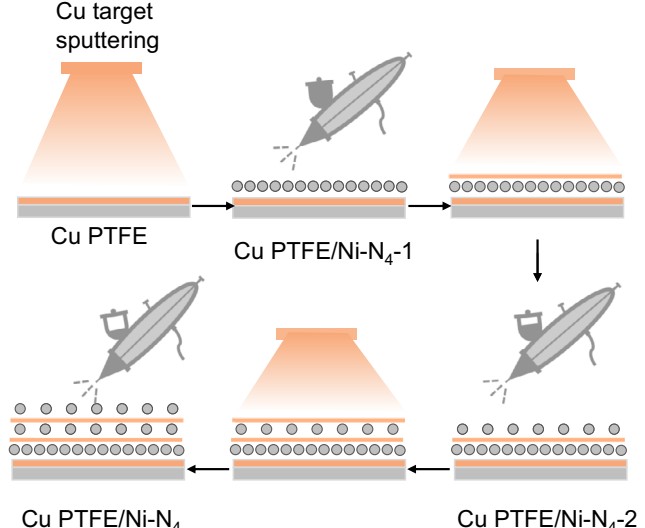

**Fig. 2 | Schematic of the process for constructing the Cu PTFE/Ni–N$_4$ composite catalysts.** Cu PTFE is fabricated by sputtering 200 nm of Cu onto a porous hydrophobic PTFE substrate. A layer of Ni–N$_4$ catalyst is then spray coated onto the Cu PTFE to yield Cu PTFE/Ni–Ni$_4$-1. Cu PTFE/Ni–Ni$_4$-2 is made by sputtering 5 nm of Cu, followed by spraying another layer of Ni–N$_4$ catalyst. This step is repeated another time to obtain the final Cu PTFE/Ni–Ni$_4$ composite catalyst.

remedied using tandem catalysis, whereby a CO-generating catalyst is located in close proximity to Cu. Cu active sites are thus exposed to a mix feed of CO$_2$/CO, which promotes the formation of C$_{2+}$ products. Although we previously reported that acidic conditions are not so favorable for tandem catalysis[40], some improvements in the C$_{2+}$ FE were still observed over the pure Cu base case. Hence, we opted to apply this strategy to increase the C$_{2+}$ product FE of our Cu PTFE catalysts.

In this work, our CO-generating catalyst of choice was a single-atom Ni catalyst[42–47], consisting of Ni–N$_4$ active sites hosted on a carbon support (see supporting information for full synthesis details). Hence, we termed this as Ni–N$_4$ based on its operating active site for electrochemical CO$_2$ to CO conversion. Figure 1b–d and Fig. S7 are representative SEM and high-resolution transmission electron microscopy (HR-TEM) images of Ni–N$_4$ with a nanosphere morphology. Aberration-corrected high-angle annular dark-field scanning transmission electron microscopy (AC-HAADF-STEM) shows dense bright spots corresponding to Ni single-atoms that are dispersed homogeneously throughout the carbon support (Fig. 1d). Energy-dispersive X-ray spectroscopy (EDS) mapping reveals uniform distribution of the elements C, N, and Ni (Fig. 1f).

XRD patterns (Fig. S8) of the Ni–N$_4$ electrocatalysts also indicate the absence of any metallic Ni phases, with only peaks corresponding to graphitic carbon. The Ni content is also quantified to be ~0.81 wt% using inductively coupled plasma atomic emission spectrometry (ICP-OES). The XPS survey spectra of Ni–N$_4$ shows the expected peaks associated with C, N, and Ni. In the Ni 2$p$ narrow scan (Fig. S9), the binding energy of Ni $_{2p3/2}$ at 854.6 eV is in the range between Ni$^0$ (853.0 eV) and Ni$^{2+}$ (855.7 eV), indicating a weakly oxidized state of Ni species in Ni–N$_4$[48]. The narrow scan N 1s XPS spectra of Ni–N$_4$ can be fitted into four characteristic peaks, among which is the pyridinic N species that can act as coordinating sites for the single Ni metal atoms[49,50].

The coordination environment of the Ni species was also investigated using X-ray absorption fine spectroscopy (XAFS). In Fig. S10, the X-ray absorption near-edge structure (XANES) spectrum of the Ni $K$-edge of Ni–N$_4$ is shown, along with standard data for Ni foil and NiO

as references. The XANES spectrum of Ni–N$_4$ falls between that of Ni foil and NiO, indicating that the valence state of the isolated Ni atoms ranges between the metallic (Ni$^0$) and oxidized (Ni$^{2+}$) states[51,52]. The extended X-ray absorption fine structure (EXAFS) spectra of Ni–N$_4$ was also analyzed using a Fourier transform (FT) k$^3$-weighted χ(k) function. The results show a significant peak at 1.31 Å, which is indicative of Ni–N coordination. In comparison, Ni foil exhibits the distinctive Ni–Ni pair at 2.15 Å, while the Ni–O interaction in NiO is evident around 1.65 Å[53]. In addition, the least-squares EXAFs curve fitting shows that Ni atoms are coordinated with four N in the first shell, indicating that the Ni species in the catalyst exists predominantly as Ni–N$_4$ (Fig. S10).

CO$_2$R was then carried out with the Ni–N$_4$ catalyst using the same flow cell system and electrolyte (0.05 M H$_2$SO$_4$ and 0.5 M K$_2$SO$_4$). From the results (Fig. S11 and Table S3), we found that these catalysts yielded a high CO FE (>90%) in an applied current density range of 100–400 mA cm$^{-2}$. This value increases gradually with higher current density, with a CO FE of ~99% at 400 mA cm$^{-2}$. Hence, these results demonstrate that our Ni–N$_4$ catalysts can effectively perform the role of CO$_2$-to-CO conversion for tandem catalysis.

Next, we built the composite catalyst (Fig. 2) by first spray coating a layer of Ni–N$_4$ catalysts (loading 0.25 mg cm$^{-2}$) onto the surface of Cu PTFE to obtain Cu PTFE/Ni–N$_4$-1. In the second step, 5 nm of Cu was sputter deposited followed by another layer of Ni–N$_4$ (loading 0.125 mg cm$^{-2}$) to form Cu PTFE/Ni–N$_4$-2. This second step was repeated another time, which yields the final Cu PTFE/Ni–N$_4$ composite catalyst. Figure 1e and Fig. S12 show representative SEM images and Fig. S13 and Fig. S14 show the XRD patterns and XPS spectra of the Cu PTFE/Ni–N$_4$ composite catalyst. The Raman spectra (Fig. S15) of Cu PTFE/Ni–N$_4$ are consistent with that of Ni–N$_4$, indicating that the coating process does not directly alter the properties of Ni–N$_4$. The double-layer capacitance of the catalyst was also determined using cyclic voltammetry (Fig. S16) and as expected, we found Cu PTFE/Ni–N$_4$ to have the largest value, indicating a higher electrochemically active surface area. Pb UPD experiments (Fig. S17) showed similar results to bare Cu PTFE, which indicates that the addition of the Ni–N$_4$ layer does not directly alter the properties of Cu. From electrochemical impedance spectroscopy (EIS) measurements (Fig. S18), we also found that Cu PTFE/Ni–N$_4$ exhibits a significantly lower charge transfer resistance (R$_{ct}$) value of 7.29 Ω compared to Ni–N$_4$ (10.13 Ω) and Cu PTFE (8.91 Ω).

CO$_2$R in acidic media was then carried out with the Cu PTFE/Ni–N$_4$ catalysts. The results at 200 mA cm$^{-2}$ (Figs. S19, S20 and Table S4-7) show significant improvements in the C$_{2+}$ FE with a value of 66.7% with the composite Cu PTFE/Ni–N$_4$ catalyst as compared to only 29.5% with bare Cu PTFE (Fig. 3a). Up to a certain limit, the C$_{2+}$ FE is also observed to increase with more layers of Cu and Ni–N$_4$ (Fig. S21 and Table S8-11), and we postulate this to be due to a higher density of interfaces between Cu and Ni–N$_4$ which are beneficial for tandem catalysis. Cu PTFE/Ni–N$_4$ was also tested at current densities of 100, 400 and 600 mA cm$^{-2}$, with the results shown in Fig. 3b. We found that the highest C$_{2+}$ FE occurs at 400 mA cm$^{-2}$, with a value of 82.4% and C$_{2+}$ partial current density of 329.7 mA cm$^{-2}$. From the results, ethylene and ethanol are the dominant C$_{2+}$ products, with a FE of 44.2% and 29.6% respectively. Notably, we find that this performance is comparable to recently reported state-of-the-art acidic CO$_2$R systems[26,29–31,33].

We also tested the long-term stability of the composite catalyst (Fig. S22) at 200 mA cm$^{-2}$ for 24 h and found no significant deterioration in performance over this period. The full cell operating voltage was also measured to be 6.5 V (Tables S6–7), which means that our system has a C$_{2+}$ full-cell EE of 14.5 %. In addition, XRD (Fig. S13), XPS (Fig. S23), and SEM (Fig. S24) characterization of Cu PTFE/Ni–N$_4$ was also carried out after the stability measurement, and we find no significant differences in the catalyst as compared to the pristine case. Cu 2$p_{3/2}$ XPS fitting curves revealed that the Cu surface was mainly in the

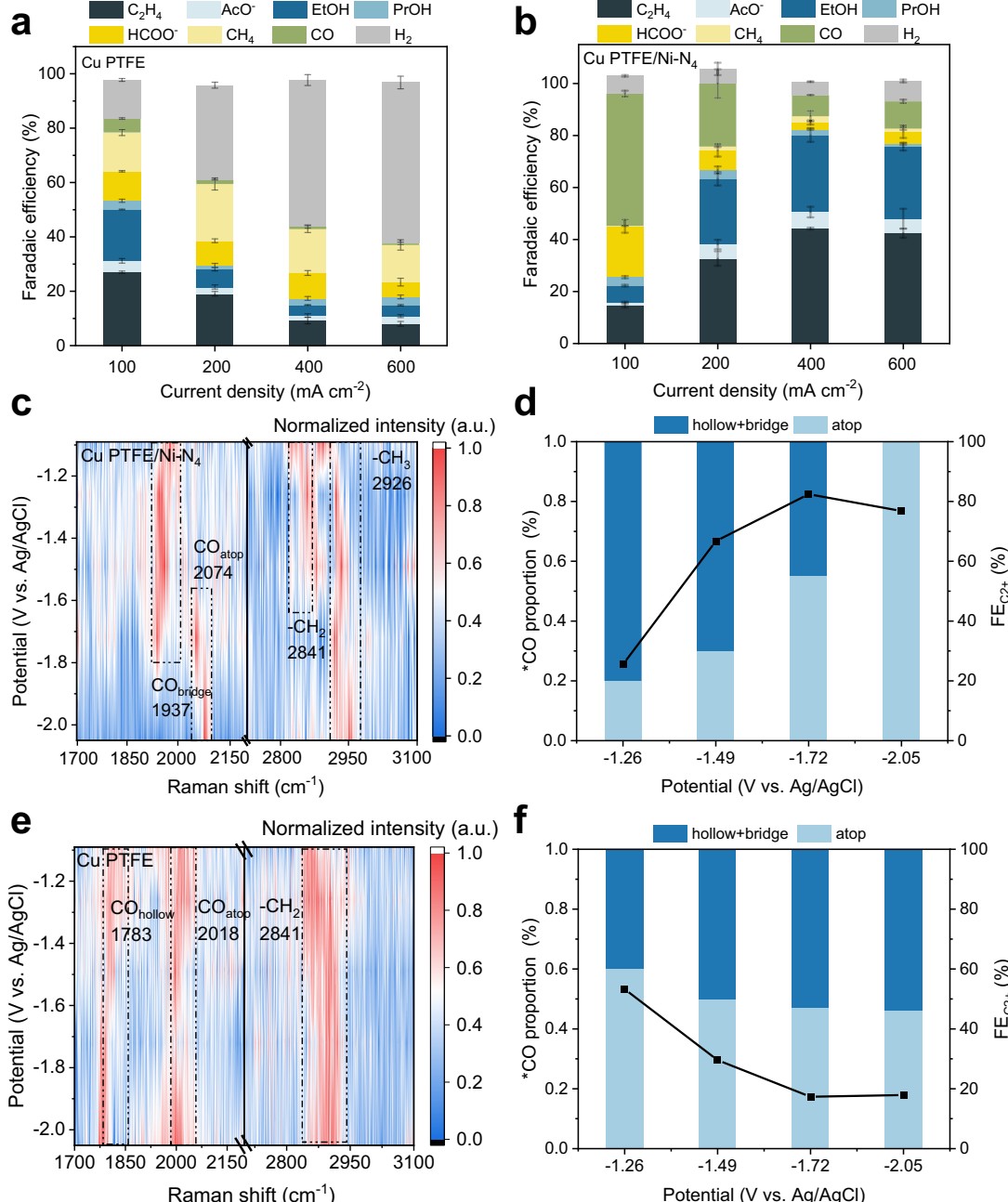

**Fig. 3 | CO₂R FE results and in situ Raman spectroscopy studies.** CO₂R FE results as a function of current density for: (**a**) Cu PTFE and (**b**) Cu PTFE/Ni−N₄ in acidic electrolyte. **c** Potential resolved in situ Raman spectroscopy of Cu PTFE/Ni−N₄ in acidic electrolyte during CO₂R. **d** Correlation between the proportion of *CO$_{atop}$ and FE towards C$_{2+}$ products for Cu PTFE/Ni−N₄ in acidic electrolyte. **e, f** Potential resolved in situ Raman spectroscopy of Cu PTFE. All the error bars represent standard deviation based on three independent samples.

oxidized state in Cu PTFE/Ni−N₄[54,55] both before and after CO₂R (Fig. S14 and Fig. S23) and is consistent with our Raman spectroscopy results (Fig. S5). This is due to oxidation of the Cu surface in ambient air and reduces to its metallic state upon application of a cathodic potential during CO₂ reduction[56,57].

In situ X-ray absorption spectroscopy (XAS) experiments were also performed on Cu PTFE/Ni−N₄ to investigate the changes of Cu valence state and structure during CO₂R (Fig. 3d and Figs. S25, S26). As soon as a cell voltage of −2.0 V vs Ag/AgCl was applied, immediate reduction of Cu oxides to metallic Cu was observed. Extended X-ray absorption fine structure (EXAFS) analysis showed that the Cu first shell coordination switched from CuO (Cu−O bond at ∼1.5 Å) to Cu−Cu (∼2.2 Å) upon application of the cathodic electrochemical potential.

Hence, these results show that metallic Cu serves as the active site for CO₂R and is consistent with previous results in the literature[28,56,57].

To understand why Cu PTFE/Ni−N₄ performs better than bare Cu PTFE, we performed in situ Raman spectroscopy measurements using a custom-made electrochemical flow cell (Fig. S27). Upon application of a potential, Raman peaks located at 1,750-2,100 cm⁻¹ (C≡O stretching of *CO) and 2700-3,100 cm⁻¹ (C–H vibration) appear[58,59]. Based on previous work by Waegele and co-workers[60], we can classify three *CO adsorption modes according to: (1) hollow-adsorbed CO at ∼1780 cm⁻¹ (*CO $_{hollow}$), (2) bridge CO at ∼1940 cm⁻¹ (*CO$_{bridge}$) and (3) low-frequency band linear CO at ∼2070 cm⁻¹ (CO$_{atop}$). A contour map of the stretching vibrations of *CO and C-H on Cu PTFE/Ni−N₄ and Cu PTFE are shown in Fig. 3c and Fig. 3e, respectively.

With Cu PTFE/Ni−N$_4$, we note the presence of peaks related to −CH$_3$ and −CH$_2$ at around 2700-3,100 cm$^{-1}$. As for *CO, we observed *CO$_{bridge}$ in the region of 1750-2,100 cm$^{-1}$, *CO$_{atop}$ centered at 1973 and 2074 cm$^{-1}$ and the absence of *CO$_{hollow}$. At the lower overpotentials, absorbed CO mainly exists as *CO$_{bridge}$ (Fig. 3d). However, at the larger overpotentials, the proportion of *CO$_{atop}$ increases at the expense of *CO$_{bridge}$. For instance, at the most negative potential (−2.05 V vs Ag/AgCl), *CO$_{bridge}$ is no longer observed and *CO$_{atop}$ becomes the only adsorbed CO species. We postulate that at the higher overpotentials, CO generation by Ni−N$_4$ becomes accelerated, which increases the CO tandem supply to the Cu sites. These findings are consistent with results by Bao and co-workers, where they found that a higher CO pressure in the feed leads to an increased *CO coverage and proportion of *CO$_{atop}$ on Cu[58]. In Fig. 3d, the highest proportion of CO$_{atop}$ was identified to be at −2.05 V vs Ag/AgCl. On the other hand, the optimal voltage for peak C$_{2+}$ FE was identified to be at −1.72 V vs Ag/AgCl. Previous work by Li et al.[59] found a volcano relationship between the C$_{2+}$ FE and the CO$_{atop}$ to CO$_{bridge}$ ratio. In our case, we postulate that a similar situation could be occurring, with an optimal CO$_{atop}$ to CO$_{bridge}$ ratio occurring at −1.72 V vs Ag/AgCl, resulting in peak C$_{2+}$ FE at this potential.

On the other hand, the in situ Raman spectroscopy results are quite different with bare Cu PTFE. For bare Cu PTFE, we do not observe peaks related to −CH$_2$ (Fig. 3e), which contrasts with our results with Cu PTFE/Ni−N$_4$ (Fig. S28 and Table S12). We also note the absence of *CO$_{bridge}$ and the presence of *CO$_{hollow}$ instead (Fig. S29 and Table S13). The proportion of *CO$_{atop}$ also decreases with increasing overpotentials, which is opposite to that with Cu PTFE/Ni−N$_4$. Hence without a tandem supply of CO, the *CO coverage and proportion of *CO$_{atop}$ on Cu is lower, which then leads to a decrease in the C$_{2+}$ product FE (Fig. 3f). Based on these results, we reason that tandem catalysis works to increase the proportion of *CO$_{atop}$ towards a more optimal CO$_{atop}$ to CO$_{bridge}$ ratio, resulting in a higher FE towards C$_{2+}$ products[59].

In addition, we conducted in situ Raman spectroscopy tests with Cu PTFE/Ni−N$_4$ to obtain an indication of the local pH of the electrode based on the HCO$_3^-$ and CO$_3^{2-}$ peaks[61]. In the voltage range of −1.09 to −2.05 V vs. Ag/AgCl, we did not observe the presence of HCO$_3^-$ or CO$_3^{2-}$ peaks in the Raman spectrum (Fig. S30a). This absence indicates that the local pH of the electrode remains acidic, since a neutral or alkaline pH is necessary for HCO$_3^-$ or CO$_3^{2-}$ to be present. Based on acid-base equilibria, we therefore deduced that the local pH should be a value of 4.5 or lower[62]. To verify these results, we repeated the same experiments, except that 1 M KOH was added to shift the bulk pH from a value of 1.7 to 2.2. In this case, we began to observe the HCO$_3^-$ and CO$_3^{2-}$ peaks in the Raman spectra (Fig. S30b). This indicates that this electrolyte can no longer maintain the local pH in an acidic range under CO$_2$R conditions.

Having established an effective electrocatalyst for CO$_2$R in acidic media, our next goal was to uncover the conditions which are beneficial towards enabling the direct conversion of CO$_2$ in flue gas. Since the thermodynamic potential of ORR is >1 V positive compared to CO$_2$R, the presence of O$_2$ impurities is the primary challenge that must be overcome[20]. We first performed ORR experiments by introducing pure O$_2$ into the gas chamber and electrolyte, using the same electrochemical cell for CO$_2$R experiments. With this, the ORR activity of Cu PTFE, Ni−N$_4$ and Cu PTFE/Ni−N$_4$ was evaluated in both alkaline (1 M KOH) and acidic media (0.05 M H$_2$SO$_4$ + 0.5 M K$_2$SO$_4$). For all three catalysts, the results (Fig. 4a and Fig. S31) indicate that ORR activity becomes significantly suppressed under acidic conditions. For example, with Cu PTFE/Ni−N$_4$ at a potential of 0.41 V vs RHE, the ORR current density decreases from −0.91 mA cm$^{-2}$ in alkaline electrolyte to −0.17 mA cm$^{-2}$ in acidic electrolyte.

Next, we performed DFT simulations to understand the suppression of ORR activity under acidic conditions. Specifically, we

investigated the different stages in the ORR reaction pathway (*OOH, *O, and *OH) on two structures: Cu (111), and Ni−N$_4$ (Figs. S34, S35). For both Ni−N$_4$ and Cu (111), the first step involving O$_2$ to *OOH has the largest uphill free energy change in the entire reaction pathway. Based on these calculations, we find that the free energy change of these steps becomes larger under acidic conditions. For instance, they increase from 1.21 eV to 1.38 eV and 0.87 eV to 1.04 eV for Ni−N$_4$ and Cu (111) respectively under acidic conditions (Fig. 4b), which are consistent with our experimental observations.

Encouraged by these findings, we performed CO$_2$R experiments using simulated flue gas (3% O$_2$ v/v, 15% CO$_2$ v/v and N$_2$ balance) with both Cu PTFE/Ni−N$_4$ and Cu PTFE in acidic and alkaline electrolyte. Because ORR yields water as the dominant product, its quantification is challenging since aqueous electrolyte is employed. Hence for all cases, we attribute the missing FE entirely towards ORR. With Cu PTFE, we find that the generation of C$_{2+}$ products is significantly suppressed in alkaline electrolyte, with a maximum FE of only 2.3% at 200 mA cm$^{-2}$, along with a 40.5% FE towards ORR (Fig. 4c and Tables S14, 15).

Once acidic electrolyte is employed, the C$_{2+}$ product FE of Cu PTFE increases substantially by ~2.5 times to a value of 5.7% at 200 mA cm$^{-2}$, together with a decrease in the ORR FE to 22.8% (Fig. S36 and Tables S16-17). Similar results are also observed with the Cu PTFE/Ni−N$_4$ composite catalyst, where ORR is suppressed and C$_{2+}$ product formation is promoted in acidic electrolyte. For instance, in alkaline electrolyte, the C$_{2+}$ product FE is 16.1% with an ORR FE of 48.2% at 300 mA cm$^{-2}$ (Fig. S37 Table S18-19). When acidic electrolyte is employed, the C$_{2+}$ product FE rises to 29.1% and the ORR FE decreases to 39.9% at 300 mA cm$^{-2}$ (Fig. S38 and Table S20-24). However, we also note that the HER FE increases slightly in acidic media. For instance, the HER FE for Cu PTFE/Ni−N$_4$ was found to be 20.6% in alkaline electrolyte (Fig. S37) and 30.6% in acidic electrolyte (Fig. S39) at a current density of 200 mA cm$^{-2}$. This is consistent with our DFT simulations (Figs. S32, S33), where we found that HER is indeed more facile under acidic conditions[63–65].

To further improve the C$_{2+}$ FE, we first replaced the 0.5 M K$_2$SO$_4$ in the electrolyte with 0.5 M Cs$_2$SO$_4$, since Cs$^+$ ions are known in the literature to be better than K$^+$ ions at promoting C-C coupling[66,67]. Employing this new acidic electrolyte formulation (Figs. S39, S40 and Tables S25-32), we observed an increase in the C$_{2+}$ product FE to 38.9% on Cu PTFE/Ni−N$_4$, with a C$_{2+}$ full-cell EE of 10.2% at 200 mA cm$^{-2}$. We then sought to further enhance the C$_{2+}$ product FE and full-cell EE by increasing the Cs$^+$ concentration of the electrolyte. This is because previous studies have indicated that an increased cation concentration can suppress HER in acidic systems[32]. Furthermore, a higher salt concentration would serve to improve the conductivity of the electrolyte and enhance the full-cell EE. Hence, by increasing the Cs$^+$ concentration from 1 M to 3 M (0.05 M H$_2$SO$_4$ + 1.5 M Cs$_2$SO$_4$), we observed a further increase in the C$_{2+}$ product FE to 46.5% and a decrease in the ORR FE to 21.0% with Cu PTFE/Ni−N$_4$ at 200 mA cm$^{-2}$ (Fig. 4c, Fig. S41 and Tables S33–40). Notably, this C$_{2+}$ product FE is ~20 times higher than Cu PTFE in alkaline electrolyte, which exemplifies the success of our electrolyte optimization and catalyst design strategies.

We also found that Cu PTFE/Ni−N$_4$ exhibits a C$_{2+}$ EE of 14.6% at 200 mA cm$^{-2}$ (Fig. S42), which was calculated based on the non-IR compensated full-cell operating voltage. Strikingly, this result is comparable to previously reported acidic systems in the literature that employ pure CO$_2$ (Fig. 4d). Finally, we operated the system for 24 h at 200 mA cm$^{-2}$ with this simulated flue gas feed, where we observed a stable full-cell operating voltage and no significant changes in the FE towards C$_{2+}$ products over this extended testing period (Fig. 4e and Table S41).

An in situ Raman spectroscopy study of Cu PTFE/Ni−N$_4$ was also performed in 0.05 M H$_2$SO$_4$ + 1.5 M Cs$_2$SO$_4$ electrolyte with the simulated flue gas feedstock. As shown in Fig. S43a, the peaks at 970 and 1040 cm$^{-1}$ represent the O−O stretching vibration of *O$_2$[68] and *OH

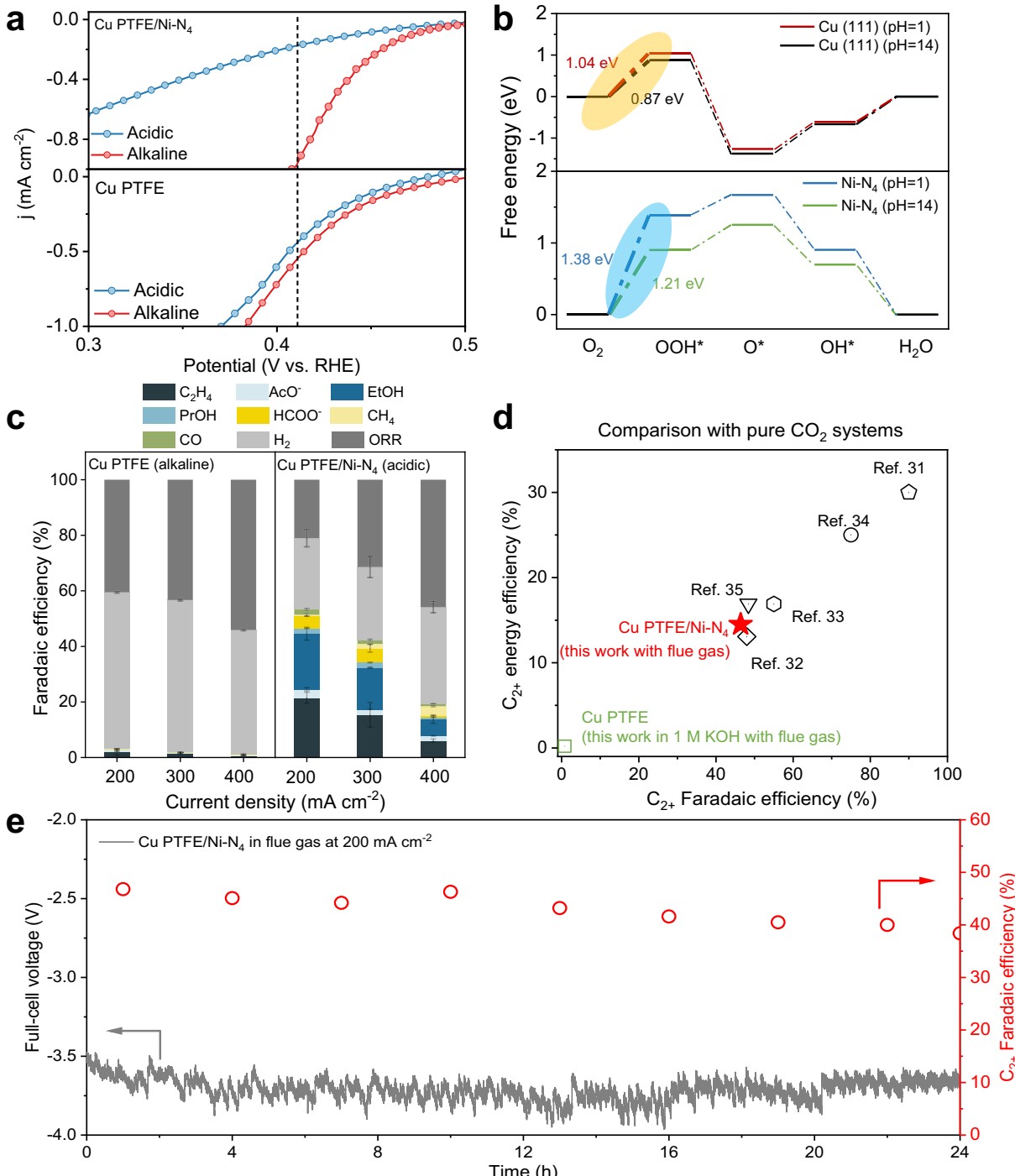

**Fig. 4 | DFT simulations of the ORR pathway and CO₂R results with simulated flue gas. a** LSV curves in pure O₂ saturated acidic and alkaline electrolyte. **b** Free energy diagrams of ORR on Ni–N₄ and Cu PTFE at 1.23 V vs. SHE. The highlighted steps show that the free energy change of the rate-determining step is increased under acidic conditions. **c** Product FE for Cu PTFE in 1 M KOH and Cu PTFE/Ni–N₄ in 0.05 M H₂SO₄ + 1.5 M Cs₂SO₄ under different current densities. **d** Comparison of

the $C_{2+}$ product full-cell EE and FE with previously reported acidic CO₂R systems[31–35]. We note that these systems run on pure CO₂ in contrast with our system which runs on simulated flue gas. **e** Full-cell voltage and $C_{2+}$ product FE of Cu PTFE/Ni–N₄ at −200 mA cm⁻² using simulated flue gas over a 24 h testing period. All the error bars represent standard deviation based on three independent samples.

species[69] respectively, which are ORR intermediates. Further decreasing the potential causes the Raman peak around 1040 cm⁻¹ to slightly weaken, while the Raman peak at 970 cm⁻¹ remains relatively unchanged regardless of the applied potential. As for the *CO population, we surprisingly only observed the presence of CO$_{atop}$ (Fig. S43b), which differs from the results with pure CO₂ feedstock (Fig. 4c). Since ORR occurs simultaneously on the same catalyst

surface, we postulate that the absence of CO$_{bridge}$ with simulated flue gas could be due to the competitive adsorption of ORR intermediates.

## Discussion

In this work, our goal was to enable direct conversion of CO₂ in simulated flue gas to $C_{2+}$ products through a combination of electrolyte selection and catalyst design. We first began by developing a Cu

PTFE/Ni–$N_4$ composite catalyst for $CO_2R$ in an acidic electrolyte. In this catalyst, Ni–$N_4$ serves to convert $CO_2$ to CO, which then transfers over to Cu active sites and boosts their $C_{2+}$ product FE. Optimization of the Ni–$N_4$ and Cu layers in this composite resulted in a catalyst that achieved a total $C_{2+}$ FE of 82.3% at 400 mA $cm^{-2}$ with pure $CO_2$ feedstock. We then tested these catalysts for ORR, where we found significant suppression of activity in acidic media for both Cu and Ni–$N_4$. This was supported by our DFT simulations where we found increases in the free energy change of the rate-determining steps for ORR on Cu and Ni–$N_4$ in acidic media. Encouraged by these results, we employed the Cu PTFE/Ni–$N_4$ catalyst for direct conversion of $CO_2$ in simulated flue gas, where we obtained a total $C_{2+}$ FE of 46.5% at a current density of 200 mA $cm^{-2}$ with an acidic electrolyte. Notably, this $C_{2+}$ FE is ~20 times higher than bare Cu PTFE with a value of only 2.3% under alkaline conditions. We also showcase stable performance for 24 h with a multicarbon ($C_{2+}$) full-cell EE of 14.6%. This result is comparable to previously reported acidic $CO_2R$ systems that employ pure $CO_2$. Importantly, our results demonstrate a potential pathway towards the design of efficient electrolyzers for direct conversion of $CO_2$ in flue gas, using simple yet effective electrolyte and catalyst design strategies.

## Methods

### Chemicals
2,4-dihydroxybenzoic acid (≥99.9%), hexamethylenetetramine (≥99.9%), Pluronic P123 (≥99.0%), sodium oleate (≥99.0%), potassium sulfate (≥99.0%), cesium sulfate (≥99.0%), perchloric acid (70%), lead (II) perchlorate trihydrate (98%), potassium hydroxide (≥85%), sulfuric acid (95.0-98.0%), and nickel (II) acetylacetonate (≥99.9%) were purchased from Sigma-Aldrich. The hydrophobic porous polytetrafluoroethylene (PTFE) substrates with 0.45 μm pore size were purchased from Beijing Zhongxingweiye Instrument Co., Ltd. Deionized water (18.2 MΩ) from an OmniaPure UltraPure Water System (Stakpure GmbH) was used for all the experiments. The Cu (99.999%) target was obtained from Kurt J. Lesker Company. Carbon dioxide (99.9%), nitrogen (99.99%), and oxygen gas (99.99%) were obtained from Air Liquide Singapore Pte. Ltd. Nafion 117 proton exchange membrane, Fumasep FAA-3-PK-130 anion exchange membrane and Ti screen mesh were purchased from Fuel Cell Store. The electrochemical flow cell and Ag/AgCl (3 M KCl) reference electrodes were purchased from Tianjin Aida Hengsheng Technology Development Co. The standard calibration gas mixtures for calibrating the gas chromatography system were obtained from Air Liquide Singapore Pte. Ltd. All the chemicals used in this work were of analytical grade and used without further purification.

### Materials characterization
Powder X-ray diffraction pattern (PXRD) was conducted on a Rigaku D/max 2500Pc X-ray powder diffractometer with monochromatized Cu Kα radiation (λ = 1.5418 Å). Scanning electron microscopy (SEM) was performed using a JEOL JSM-7610F field scanning electron microscope. Transmission electron microscopy (TEM) was carried out using a JEOL JEM-2100F field emission electron microscope working at 200 kV. Single atoms were characterized and identified using high-angle annular dark-field scanning transmission electron microscopy (HAADF STEM) with a FEI Themis Z scanning/transmission electron microscope operated at 300 kV, equipped with a probe spherical aberration corrector. X-ray photoelectron spectroscopy (XPS) was collected on a Kratos AXIS Supra+ spectrometer equipped with a monochromatized Al Kα X-ray source and a concentric hemispherical analyser. The survey scan was carried out using an emission current of 15 mA, pass energy of 160 eV and step size of 1 eV. The narrow scan was performed using an emission current of 15 mA, pass energy of 20 eV and step size of 0.1 eV. Ex-situ XAS experiments were performed at the XAFCA beamline of the Singapore Synchrotron Light Source. In situ

XAS measurements were performed at the Australian Synchrotron Facility at ANSTO. $CO_2$ reduction gas products were analyzed using an Agilent 8600 gas chromatography system equipped with a thermal conductivity detector and a flame ionization detector. Liquid products were quantified with high-performance liquid chromatography (Agilent 1260 Infinity II HPLC). The HPLC was equipped with a refractive index and UV detector. The eluent used was 1 mM $H_2SO_4$, with Aminex HPX-87H columns from Bio-Rad laboratories.

### Preparation of Ni–$N_4$ catalysts
The first step involved the synthesis of hollow polymer spheres, which was based on a previously reported method with some modifications. In a typical procedure, 90 mg 2,4-dihydroxybenzoic acid and 184.8 mg hexamethylenetetramine were dissolved in 60 ml deionized water. To this solution, 20 ml of another solution containing 30 mg Pluronic P123 and 72.96 mg sodium oleate was added under slow stirring. After slowly stirring for 10 min, the mixed solution was transferred into a 100 ml Teflon-lined stainless-steel autoclave and heated to a temperature of 160 °C for 2 h in a heating oven. After the reaction was complete, the autoclave was left to cool to room temperature. The hollow polymer spheres were then collected by centrifugation, washed three times with deionized water and ethanol and finally dried at 60 °C in a vacuum oven. 0.2 g of the hollow polymer spheres was then dispersed in 5 ml of ethanol. To this solution, 5 mL of another ethanol solution containing 7.0 mg Ni(acac)$_2$ was added under stirring. The resulting mixture was then stirred at 80 °C until all the ethanol was evaporated. After that, the resulting mixture and 4.0 g dicyandiamide were separately placed in two alumina combustion boat located at the down-stream and up-stream direction in a tube furnace, respectively. The tube furnace was heated to 900 °C with a heating rate of 5 °C/min under flowing nitrogen gas (10 mL/min) and held at that temperature for two hours. After cooling to room temperature, the Ni–$N_4$ catalysts were obtained.

### Preparation of Cu PTFE
This was prepared by using a magnetron sputtering system (Cello Ohmiker-30CSL) to coat the porous hydrophobic PTFE membrane with 200 nm of Cu. A radio frequency (RF) power supply was used, and the sputtering power was controlled such that the total deposition rate was fixed at 2 Å/s.

### Preparation of Cu PTFE/Ni–$N_4$ composite catalysts
To prepare the composite catalysts, 8 mg of Ni–$N_4$ catalyst was added into a mixed solution of 1.92 mL isopropanol and 80 μL Nafion solution. The resulting mixed solution was then ultrasonically treated for 2 h to form a homogeneous ink. After that, the Ni–$N_4$ catalyst ink (1 ml) was sprayed onto the Cu PTFE (4 cm by 4 cm) using an air brush (loading 0.25 mg $cm^{-2}$) and this electrode was named Cu PTFE/Ni–$N_4$-1. Cu PTFE/Ni–$N_4$-2 is prepared by sputtering of 5 nm of Cu on Cu PTFE/Ni–$N_4$-1, followed by spraying 0.5 ml of the Ni–$N_4$ catalyst ink (loading 0.125 mg $cm^{-2}$). Cu PTFE/Ni–$N_4$ is produced by magnetron sputtering of another 5 nm of Cu onto Cu PTFE/Ni–$N_4$-2, followed by spraying 0.5 ml of the Ni–$N_4$ catalyst ink (loading 0.125 mg $cm^{-2}$). Cu PTFE/Ni–$N_4$-4 is prepared by sputtering of 5 nm of Cu onto Cu PTFE/Ni–$N_4$, followed by spraying 0.5 ml of the Ni–$N_4$ catalyst ink (loading 0.125 mg $cm^{-2}$).

### In situ Raman spectroscopy
In situ Raman spectroscopy was carried out with a Horiba LabRam Odyssey Nano Raman Spectrometer system. Measurements were performed using a custom-made in situ electrochemical flow cell, with a gas chamber at the backside of the gas diffusion electrode for continuous $CO_2$ flow. An Olympus N2667700 water immersion objective was dipped into the electrolyte to collect the Raman spectra. IrO$_x$ coated Ti mesh were used as the counter electrode in acidic

electrolyte. The Ag/AgCl (3 M KCl) reference electrode was used for all experiments.

## Electrochemical measurements

All electrochemical measurements were carried out using an Autolab PGSTAT204 potentiostat. $CO_2R$ experiments were conducted in a gas diffusion electrode electrochemical flow cell system with an electrode exposed area of 1 cm$^2$. High-purity $CO_2$ gas flowed at a rate of 30 sccm behind the cathode GDL controlled by a mass flow controller (MC-2000SCCM-D/5 M, Alicat Scientific). The flow cells were assembled with $IrO_x$/Ti mesh as the anode, Ag/AgCl as the reference electrode, and a Nafion exchange membrane (Nafion 117; size: 2.5 cm by 2.5 cm; thickness: 0.18 mm) to separate the cathode and anode chambers. The $IrO_x$/Ti mesh electrode was prepared using a dip coating and thermal decomposition method, according to methods described by Luc et al.[70]. The Nafion proton exchange membrane was activated in 5 wt.% $H_2SO_4$ at 80 °C for 2 h before use.

$CO_2R$ with simulated flue gas (3% $O_2$ v/v, 15% $CO_2$ v/v and $N_2$ balance) was performed using the same electrochemical cell. For these experiments, the missing FE was assumed to be entirely attributed to ORR, due to the inability to quantify the amount of product ($H_2O$) generated. To evaluate the ORR activity of our catalysts, we flowed pure $O_2$ into the gas chamber as well as the electrolyte of the electrochemical cell. These ORR tests were conducted in either 1 M KOH (alkaline) or 0.05 M $H_2SO_4$ + 0.5 M $K_2SO_4$ (acidic) electrolyte. To saturate the electrolyte with $O_2$, the electrolyte was bubbled with $O_2$ for 20 min prior to each experiment. Linear sweep voltammetry (LSV) tests were conducted at a sweep rate of 5 mV s$^{-1}$. Before the actual LSV measurement, CV cycling was conducted to obtain stable curves.

For gas product quantification, 1 mL of the gas exiting the electrochemical cell was injected into the gas chromatograph using a gas-tight syringe. The Faradaic efficiency (FE) of the gas products were calculated based on the following equation:

$$\text{Faradaic efficiency}(\%) = N \times F \times v \times c / (i \times V_m) \tag{1}$$

where $N$ is the number of electrons transferred, $F$ is the Faraday constant, $v$ is the gas flow rate, $c$ is the concentration of the detected gas product in ppm, $i$ is the total current and $V_m$ is the unit molar volume of gas. The gas flow rate was measured at the outlet of the electrochemical cell using a bubble flow meter.

$FE$ of liquid products was determined as below:

$$FE = \frac{Q_{\text{liquid}}}{Q_{\text{total}}} * 100\% = \frac{nNF}{Jt} * 100\% \tag{2}$$

where $n$ is the moles of liquid product in the cathodic compartment, $N$ is the electron transfer number, $F = 96{,}485$ C mol$^{-1}$, t is the reaction time, $J$ is the recorded current.

The energy efficiency (EE) for the formation of $C_{2+}$ products is calculated as follows:

$$EE = \frac{\sum_i^n FE_i * \left(1.23 - E_i^0\right)}{V_{fullcell}} \tag{3}$$

where $V_{fullcell}$ is the full cell voltage applied in the experiment (without ohmic loss correction) and $FE_i$ is the measured Faradaic efficiency of each product. $E_i^0$ is the standard reduction potential of each product (ethylene: 0.08 V) (ethanol: 0.09 V) (acetate: -0.26 V) (propanol: 0.21 V).

## Density functional theory simulations

The projected augmented wave (PAW) approach[71] and the generalized gradient approximation (GGA) of Perdew, Burke and Ernzerhof (PBE)[72] exchange-correlation functional (widely accepted for catalysis calculations) were employed in the Vienna ab initio Simulation Package (VASP)[73] to perform all the plane wave density functional theory (DFT) simulations. To simulate the properties of the actual bulk material and surface, we employed $4 \times 4 \times 1$ Cu (111) slab in a vacuum. The top two layers of Cu (111) were allowed to move freely due to their interaction with the adsorbates, while the remaining layers were held fixed in their optimized crystalline positions. We employed a graphene supercell with surface periodicity of $4 \times 4$ graphene supercell to simulate adjacent regions of Ni–$N_4$. A vacuum distance of 18 Å was introduced in the z-direction. We used a cut-off energy of 400 eV for the plane wave basis sets and a $4 \times 4 \times 1$ Γ centered Monkhorst-Pack mesh for k-point sampling in the first Brillouin zone. The convergence criteria for energy and forces were set at $1 \times 10^{-5}$ eV and 0.01 eV Å$^{-1}$, respectively[74]. The computational hydrogen electrode was utilized to obtain free energies for each state as done in ref. [75]. We calculated the reaction free energies of oxygen reduction reaction (ORR) on Cu (111) and Ni–$N_4$ surfaces, considering their dependence on the potential. For evaluating the $H_2O$ dissociation energy barrier, the transitional state was located using the Nudged Elastic Band method. The free energy of adsorbed H ($\Delta G_H$) on surfaces is expressed as Eq. (4):

$$\Delta G_H = \Delta E_H + \Delta E_{ZPE} - T\Delta S \tag{4}$$

where $\Delta E_H$ is the hydrogen adsorption energy, $\Delta E_{ZPE}$ and $\Delta S$ are the zero-point energy difference and the entropy difference between the adsorbed state and the gas phase, respectively, and $T$ is the system temperature (298.15 K).

## Data availability

The authors declare that the data supporting the findings of this study are available within the paper and its Supplementary Information files. Should any raw data files be needed in another format they are available from the corresponding author upon request.

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

## Acknowledgements

Y.L. and Jia Z. acknowledge support and funding from the A*STAR (Agency for Science, Technology, and Research) under its LCERFI program (Award No: U2102d2002). Y.L. acknowledges support and funding from the NRF Fellowship (Award No: NRF-NRFF14-2022-0003). We acknowledge Dr. Bernt Johannessen and the use of the Australian Synchrotron Facility at the Australian Nuclear Science and Technology Organisation (ANSTO) for the collection of the in situ XAS data used in this work. We acknowledge the use of the XAFCA beamline of the Singapore Synchrotron Light Source (SSLS) for the collection of the ex-situ XAS data used in this work.

## Author contributions

Y.L. supervised the project. Y.L. and M.W. conceived the idea and designed the experiments. M.W. and B.W. carried out all the experimental work. M.W. and Jia Z. performed and supervised the computational work, respectively. M.W. and Jiguang Z. performed the in situ Raman spectroscopy experiments and catalyst synthesis. M.W., B.W., and Q.Y. carried out the in situ XAS experiments. S.X. carried out the ex-situ XAS experiments. B.W. prepared the $IrO_x$-coated Ti mesh electrodes and Ni single atom catalysts. M.Z. carried out the XPS measurements. Jiguang Z. carried out the XRD measurements and analysis. N.L. prepared the Cu PTFE catalysts. Z.M. and W.R.L. contributed to data analysis and manuscript editing. Y.L. and M.W. co-wrote the manuscript. All authors discussed the results and assisted during the manuscript preparation.

## Competing interests

The authors declare no competing interests.
