## [Peer Review File · Nature Communications]

REVIEWER COMMENTS

Reviewer #1 (Remarks to the Author):

In this work, the authors reported that direct conversion of CO₂ in simulated flue gas to C₂⁺ products with Cu PTFE/Ni-N₄ composite catalyst, where obtained a total C₂⁺ FE of 46.5% at a current density of 200 mA cm⁻² with acidic electrolyte for direct conversion of CO₂ in simulated flue gas. As we all know, C₂⁺ products will be produced by copper catalyst on electrochemical CO₂ reduction reaction. In the other hand, many works has been reported that CO formation with Ni-based catalyst on electrochemical CO₂ reduction reaction (Inorg. Chem. Front., 2019,6, 1729-1734; Angew. Chem. Int. Ed. 2020, 59, 4043-4050; Dalton Trans., 2023, 52, 928-935; Appl. Catal. B 2020, 271, 118929). In addition, tandem catalysis on electrochemical CO₂ reduction reaction (from CO₂ to CO, and then to C₂⁺) by different catalysts or catalytic sites have also been reported by many works (J. Energy Chem. 2022, 70, 219-223; Nano Res. 2021, 14, 4471-4486; Angew. Chem. Int. Ed. 2021, 60, 25485-25492), including the work from the same group (Angew. Chem. Int. Ed. 2023, 62, e202308782). Only changing electrocatalyst (Cu-PTFE and Ni-N-C) is not novel in the same tandem reaction condition. No unique scientific insights were provided in this work. In general, the enhanced catalytic performance by constructing the Cu PTFE/Ni-Ni₄ composite catalysts in this manuscript is within expected. Only simple combination of two well-known electrocatalysts does not attach the level of this strict scientific journal.

In addition, the authors claim that the work enables oxygen-tolerant production of C₂⁺ products during electrochemical CO₂ reduction reaction. The presence of O₂ impurities can result in ORR reaction. However, this conclusion is untenable. Many works by O₂-containing CO₂ reactant has been developed on electrochemical CO₂ reduction reaction (Nat. Commun. 2020, 11, 3844; Sci. Bull. 2019, 64, 1890-1895; Energy Environ. Sci. 2020, 13, 554-561; Angew. Chem. Int. Ed. 2020, 59, 10918-10923). Furthermore, the electrochemical CO₂ reduction reaction in a membrane-free cell has been reported (Front. Chem. 10, 915759), which was the similar reaction condition that oxygen containing. In some catalysts, the performance with O₂-containing CO₂ was better than pure CO₂. However, most of these relative works were not cited and discussed in this manuscript.

About reaction mechanism analysis, the authors performed DFT simulations to understand the suppression of ORR activity under acidic conditions. However, the reaction mechanism about electrochemical CO₂ reduction reaction should be investigated by DFT computations, which is more key section.

Therefore, this work lacks sufficient novelty to publish for the strict scientific level journal Nature Communications.

Reviewer #2 (Remarks to the Author):

The manuscript by Lum et. al. reported selective and stable electrochemical CO₂ reduction (CO₂R) from flue gas in acid. The idea of utilizing flue gas is more challenging yet practical for the commercialization of this technology. The authors designed tandem catalysts, Cu PTFE/Ni-N₄, to achieve FE(C₂⁺) of 46.5% at 200 mA/cm². Various techniques, including in-situ Raman and XAS, were used to characterize the

mechanism and active sites. The manuscript is well-written, and the experiments are carefully designed. I, thus, recommend publishing after minor revision after addressing the following questions:

1. Acidic electrolyte is selected by the authors at the beginning of the study. It is known that CO₂RR has a much higher activity in base than acid. Can the authors explain why acid is selected for CO₂RR? Besides, the solubility of K₂SO₄ is limited. This might be the reason for the large full-cell voltage (6.5 V). Did the authors try other salts with higher solubility?
2. The in-situ Raman studies are very informative to obtain insights into the reaction mechanism. CO(atop) is more active than CO(hollow+bridge) for C₂⁺ formation. In Fig 2d, the highest portion of CO_{atop} was found at -2.05V, while the faradaic efficiency of C₂⁺ peaked at -1.72 V. Did the authors consider other reasons besides CO adsorption for the selectivity of C₂⁺ formation? In addition, what is the influence on the surface CO population when oxygen is introduced, as is the case for flue gas?
3. The Cu PTFE/Ni-N₄ electrode is fabricated through a layer-by-layer strategy. Will the active site of Ni be covered by Cu, especially when the concentration of Ni is very low? Is there any systematic design strategy to ensure the exposure of the Ni active sites to proceed for tandem catalysis as suggested? In Fig 3.a, the Cu-PTFE/Ni-N₄ showed a higher ORR activity than Cu-PTFE alone. What is the rationale for choosing Ni-N₄? Did the authors consider choosing other catalysts with poorer ORR activity?

Minor comments:

Typos in Line 222. I think the authors meant Fig 2d rather than Fig 2f.

Reviewer #3 (Remarks to the Author):

In this manuscript, the authors describe a strategy for the generation of multicarbon products from simulated flue gas, where acidic electrolyte was used to significantly suppress ORR on a Cu PTFE/Ni-N₄ electrode. The authors achieve FEs toward multicarbon products of 82.4 % and 46.5% in pure CO₂ gas and in simulated flue gas respectively. This was supported theoretically using DFT simulations where they found increases in the free energy change of the rate-determining steps for ORR on Cu and Ni-N₄ in acidic media. This manuscript is suitable for publication in Nature Communications after appropriate revision, with detailed comments as given below:

1. More statistical analysis of the Ni-N₄ size should be provided, which in the ideal case should yield a distribution close to a Gaussian shape.
2. More information should be provided regarding the computational methods, such as why the top two layers of Cu (111) were chosen to move freely, the criteria for selecting K-points, and whether spin polarization was considered.
3. Did the authors test the performance of Cu PTFE in 1 M Cs⁺ and 3 M Cs⁺ solutions for CO₂R with flue gas?
4. In Figure S20, the authors claim that the CO₂R performance of Cu PTFE/Ni-N₄ surpasses that of Cu PTFE/Ni-N₄-1 and Cu PTFE/Ni-N₄-2. What will happen to the performance if more layers are added to the catalyst system?

5. Please elucidate the procedure employed by the authors to determine the area/proportion of *CO_{hollow}, *CO_{bridge}, and *CO_{atop}, as well as how they validated the ratio of *CO_(hollow + bridge) to *CO_{atop}? Is this method an original development, or has it been previously established? If this is not a new method, kindly provide additional details and cite the relevant research articles.

6. Authors should check through the manuscript for mistakes, for example there are some errors in subscripts/superscripts.

Reviewer #4 (Remarks to the Author):

The manuscript entitled "Acidic media enables Oxygen-tolerant Electrosynthesis of Multicarbon Products from Simulated Flue Gas" describes the discovery that the choice of electrolyte and catalyst design can be used to enable oxygen-tolerant production of C₂₊ products in simulated flue gas. The work presented in the manuscript has some novelties and the idea is interesting and relevant to the needs of the current CO₂ reduction experiments.

The manuscript is well organized, however, there are still some issues that need to be addressed.

Subject comments

1. In the abstract and introduction section, abbreviations Ni-N₄ need to be explained.
2. Although the author claims the catholyte is acidic (0.05M H₂SO₄ + 0.5 M K₂SO₄ or 1.5 M Cs₂SO₄), it would be nice to know about the operating pH conditions throughout the experiment.
3. The XPS spectrum in Fig S4, Fig S14 should be assigned. The survey spectrum shows some other peaks that should be identified. The high-resolution spectrum needs to be fitted properly to find out the oxidation states.
4. In Fig S6, and Fig 2 why the overall FE is not 100%, if it's not 100%, does the author expect any other products? Then in Figure S11, shows that the overall FE is over 100%. Why those discrepancies?
5. The author produced the EIS spectrum in Figure S18, but there is no significant change in the R_{ct} value of the three catalysts to claim that the Ni-N₄ catalyst possesses faster kinetics. Thus, the claim is ambiguous.
6. The authors performed DFT simulations on ORR activity, however in acidic conditions a DFT calculation about CO₂RR with hydrogen evolution path will give more insights.
7. What is the practical relevance of this single atom catalyst? How feasible is it to upscale it and bring to practical use beyond the lab scale study?
8. What about testing with real flue gas which will give a more realistic case?

We thank the editor for handling our manuscript and the reviewers for their comments, which we have used to improve the quality of the work. Changes within the manuscript and supporting information are highlighted in yellow. Below is a response to the reviewer comments, which are written in blue font.

Reviewer #1 (Remarks to the Author):

In this work, the authors reported that direct conversion of CO₂ in simulated flue gas to C₂⁺ products with Cu PTFE/Ni-N₄ composite catalyst, where obtained a total C₂⁺ FE of 46.5% at a current density of 200 mA cm⁻² with acidic electrolyte for direct conversion of CO₂ in simulated flue gas. As we all know, C₂⁺ products will be produced by copper catalyst on electrochemical CO₂ reduction reaction. In the other hand, many works has been reported that CO formation with Ni-based catalyst on electrochemical CO₂ reduction reaction (Inorg. Chem. Front., 2019,6, 1729-1734; Angew. Chem. Int. Ed. 2020, 59, 4043-4050; Dalton Trans., 2023, 52, 928-935; Appl. Catal. B 2020, 271, 118929). In addition, tandem catalysis on electrochemical CO₂ reduction reaction (from CO₂ to CO, and then to C₂⁺) by different catalysts or catalytic sites have also been reported by many works (J. Energy Chem. 2022, 70, 219-223; Nano Res. 2021, 14, 4471-4486; Angew. Chem. Int. Ed. 2021, 60, 25485-25492), including the work from the same group (Angew. Chem. Int. Ed. 2023, 62, e202308782). Only changing electrocatalyst (Cu-PTFE and Ni-N-C) is not novel in the same tandem reaction condition. No unique scientific insights were provided in this work. In general, the enhanced catalytic performance by constructing the Cu PTFE/Ni-N₄ composite catalysts in this manuscript is within expected. Only simple combination of two well-known electrocatalysts does not attach the level of this strict scientific journal.

Response

We are thankful to Reviewer #1 for taking the time to evaluate our manuscript.

We would like to emphasize that the main focus and novelty of the work is on identifying an electrolyte composition and electrocatalyst system combination that can enable the generation of multicarbon (C₂⁺) products directly from simulated flue gas. In this process, we needed to solve the two main challenges involved, which is the low CO₂ concentration (15%) and the competing oxygen reduction reaction. For the first time, we have successfully demonstrated in this work that acidic electrolyte in combination with a Cu PTFE/Ni-N₄ tandem electrocatalyst system can solve these challenges. This is the main novelty and focus of this work.

We also emphasize that the tandem electrocatalyst design for conventional electrochemical CO₂ reduction with pure CO₂ feedstock is not the main focus or novelty of this work. As the reviewer correctly states, there has already been a significant amount of literature on such tandem electrocatalyst systems. However, we note that all these studies were conducted in alkaline and neutral media. Therefore, we did not know beforehand what kind of tandem electrocatalyst configuration would exhibit high FE towards C₂⁺ products in acidic media.

Although the reviewer mentions that “*the enhanced catalytic performance by constructing the Cu PTFE/Ni-N₄ composite catalysts in this manuscript is within expected*”, we found this not to be true. In our early experimental trials, we empirically found that quite a few of the tandem electrocatalyst systems that work well in alkaline/neutral media do not actually perform very well in acidic media. This issue is not previously well-known and hence is not something that can be said to be “*within expected*”. This is the main reason why it was necessary in the first part of our manuscript to first verify that the tandem Cu PTFE/Ni-N₄ electrocatalyst system

that we designed can indeed exhibit a high FE towards C_{2+} products in acidic media using pure CO_2 feedstock. In fact our C_{2+} FE of 82.4% is quite comparable to reported state-of-the-art acidic electrochemical CO_2 reduction systems using pure CO_2 feedstock.

Only after we had developed this electrocatalyst system, could we then move on to the next step, which is the main focus and novelty of this work, where we successfully demonstrated the use of this electrocatalyst system in combination with acidic electrolyte to generate C_{2+} products from simulated flue gas with FE of 46.5% at 200 mA cm^{-2} .

However, we acknowledge that our electrocatalyst design takes inspiration from these prior works that the reviewer has mentioned, using these key ideas to design an efficient electrocatalyst for a different application: which is the direct conversion of CO_2 in simulated flue gas to C_{2+} products. Hence, we have now cited all these relevant publications in our manuscript.

In addition, the authors claim that the work enables oxygen-tolerant production of C_{2+} products during electrochemical CO_2 reduction reaction. The presence of O_2 impurities can result in ORR reaction. However, this conclusion is untenable. Many works by O_2 -containing CO_2 reactant has been developed on electrochemical CO_2 reduction reaction (Nat. Commun. 2020, 11, 3844; Sci. Bull. 2019, 64, 1890-1895; Energy Environ. Sci. 2020, 13, 554-561; Angew. Chem. Int. Ed. 2020, 59, 10918-10923). Furthermore, the electrochemical CO_2 reduction reaction in a membrane-free cell has been reported (Front. Chem. 10, 915759), which was the similar reaction condition that oxygen containing. In some catalysts, the performance with O_2 -containing CO_2 was better than pure CO_2 . However, most of these relative works were not cited and discussed in this manuscript.

Response

Figure R1 | Total current densities and Faradaic efficiencies of the polycrystalline Cu powder electrode measured at (a) pure CO_2 , (b) 90% CO_2 + 10% O_2 , (c) 80% CO_2 + 20% O_2 . These data are taken from work by He et al. (Nat. Commun. 2020, 11, 3844).

The work by He et al. (Nat. Commun. 2020, 11, 3844) found that the partial current density to C_{2+} products could be increased in CO_2/O_2 mixtures as compared to pure CO_2 alone, especially at the lower overpotentials. However, it is important to note that as a result of the O_2 addition, even though the C_{2+} partial current density increases, the total FE actually decreases significantly. This is because a significant amount of current becomes consumed by ORR. For convenience, we have reproduced their results in Fig. R1 above. We observe that with addition of O_2 , a significant portion of the current actually becomes consumed by the ORR, such that

the FE towards CO₂RR products becomes very small. This observation is also consistent with our result in alkaline media (Fig. 3c), where we found that a significant portion of the current is consumed by ORR and very little FE towards CO₂RR products. Importantly, we show in our work that acidic electrolyte can suppress the ORR and allow for reasonable FE values towards C₂₊ products. We have now cited and discussed this work in our manuscript.

The work by Lu et al. (*Sci. Bull.* 2019, 64, 1890-1895) was already cited and discussed in the introduction of our manuscript. The work by Li et al., (*Angew. Chem. Int. Ed.* 2020, 59, 10918-10923) is now cited and discussed in our manuscript. In both works, they utilized a selective CO₂ mass transport strategy by coating their cobalt phthalocyanine or Sn catalysts with a selectively permeable polymer with a high CO₂/O₂ selectivity. As a result, they achieved a 75.9% Faradaic efficiency (FE) for CO production when a feedstock containing 5% O₂ and 95% CO₂ was employed. For the Sn catalyst, they generated formate with nearly 100 % selectivity and a current density of 56.7 mA cm⁻² with a feedstock of 95% CO₂ and 5% O₂. Importantly however, they did not demonstrate if their strategy could be applied to the generation of C₂₊ products on Cu. Furthermore, in our case the CO₂ concentration we used was much lower at only 15%. We also note that it very important to develop other strategies in addition to selective CO₂ mass transport that could be simple yet effective, which we successfully demonstrated our strategy involving electrolyte selection and electrocatalyst design. This is because in future work, our strategy could be potentially combined with the selectively permeable polymer strategy to yield even better performance results.

The work by Xu et al. (*Energy Environ. Sci.* 2020, 13, 554-561) was already cited and discussed in our manuscript. In this work, they also employed a selective CO₂ mass transport strategy by using ionomer coatings on their Cu catalysts which selectively slowed down O₂ mass transport. In addition to the points raised in the previous paragraph, we note that this required high pressure conditions of 10 bar to obtain reasonable FEs to C₂₊ products. In our work, high pressure conditions are not required and ambient pressure was sufficient.

The work by Tian et al. (*Front. Chem.* 10, 915759) focused mainly on the design of CuO/CeO₂ nanocomposites for selective electrochemical CO₂ reduction to C₂₊ products. The authors explain that the addition of CeO₂ to CuO helps to stabilize Cu⁺ sites that promote C₂₊ product formation. In their work, pure CO₂ feedstock was continuously bubbled into the electrolyte to keep the electrolyte saturated with CO₂ in a membrane-free electrochemical cell. However, we note that O₂ tolerance was not the main focus of their work. Rather the design of the CuO/CeO₂ catalyst was the main focus. Hence, the authors did not perform a systematic study on O₂ tolerance. In their system, it seemed that the O₂ generated at the anode did not affect their CO₂ reduction at the cathode. We reason that this is likely because: (1) pure CO₂ is continuously sparged into the electrolyte, which will remove any O₂ generated from the anode. (2) the current densities applied are low (5-10 mA cm⁻²), which means that any O₂ generated might be quite little. (3) the solubility of O₂ in water is very low and is ~26 times lower than CO₂. As a result of these factors, there might be only a very small amount of O₂ dissolved in their electrolyte. We have now cited this work in our manuscript.

About reaction mechanism analysis, the authors performed DFT simulations to understand the suppression of ORR activity under acidic conditions. However, the reaction mechanism about electrochemical CO₂ reduction reaction should be investigated by DFT computations, which is more key section.

Response

There is already a significant amount of published literature, where the electrochemical CO₂ reduction reaction pathway was investigated using DFT simulations on Cu surfaces and Cu-based tandem systems. For instance, (*ACS Catal.*, 2021, 11(8), 4456–4463), (*Nat. Catal.*, 2020, 3, 75–82), (*ACS Catal.*, 2020, 10(7), 4059–4069), (*ACS Catal.*, 2018, 8(2), 1490–1499), (*Nat. Commun.*, 2022, 13, 1399), (*Angew. Chem. Int. Ed.*, 2013, 52, 7282-7285) and (*Proc. Natl. Acad. Sci. U. S. A.*, 2017, 114, 1795-1800). Hence, we believe that we cannot contribute any new findings or discoveries by simply repeating the same calculations. Nor can such calculations directly support or help our manuscript.

We emphasize that one focus and novelty of our work is that the ORR activity can be suppressed under acidic conditions, which can enable the generation of C₂₊ products from simulated flue gas. This is the reason why we were motivated to perform DFT simulations to understand this suppression of ORR activity.

Therefore, this work lacks sufficient novelty to publish for the strict scientific level journal *Nature Communications*.

Response

Once again, we would like to sincerely thank Reviewer #1 for reviewing our manuscript. We are also grateful for these important discussion points that have been raised. We hope that we have adequately clarified the main novelty points of our manuscript and improve their opinion of our work.

Reviewer #2 (Remarks to the Author):

The manuscript by Lum et. al. reported selective and stable electrochemical CO₂ reduction (CO₂R) from flue gas in acid. The idea of utilizing flue gas is more challenging yet practical for the commercialization of this technology. The authors designed tandem catalysts, Cu PTFE/Ni-N₄, to achieve FE(C₂₊) of 46.5% at 200 mA/cm². Various techniques, including in-situ Raman and XAS, were used to characterize the mechanism and active sites. The manuscript is well-written, and the experiments are carefully designed. I, thus, recommend publishing after minor revision after addressing the following questions:

Response

We thank Reviewer #2 for their positive evaluation of our manuscript and thoughtful suggestions that have helped us improve our work.

1. Acidic electrolyte is selected by the authors at the beginning of the study. It is known that CO₂RR has a much higher activity in base than acid. Can the authors explain why acid is selected for CO₂RR? Besides, the solubility of K₂SO₄ is limited. This might be the reason for the large full-cell voltage (6.5 V). Did the authors try other salts with higher solubility?

Response

The main goal of the project was to figure out the optimal electrolyte conditions and electrocatalyst design that would enable suppression of ORR and formation of C₂₊ products from flue gas feedstock. Hence, when we started the project, we first experimentally screened alkaline, neutral and acidic conditions for direct conversion of simulated flue gas to C₂₊ products. Through this, we found that acidic conditions allowed Cu PTFE electrodes to produce the highest C₂₊ FE and better suppression of ORR. Therefore, although the activity in base is indeed higher under conventional pure CO₂ feedstock conditions, this is not the case when flue gas is instead used as the feedstock. Hence, this is the reason why we selected acidic electrolyte at the beginning of the study. As for the exact acidic electrolyte formulation, this was based on a paper by Xie et al. (*Nat. Catal.*, 2022, 5, 564–570), which provided us with important insights.

In addition, to explore the effect of K₂SO₄ solubility on the full cell voltage, we replaced this with Cs₂SO₄ which has a higher solubility. Hence, we conducted CO₂R tests using 0.05 M H₂SO₄ + 1.5 M Cs₂SO₄ as the acidic electrolyte formulation with pure CO₂ feedstock. The C₂₊ FE achieved a value of 87.3% at a cathodic current density of 400 mA cm⁻². As correctly highlighted by the reviewer, the full-cell voltage at 400 mA cm⁻² can indeed be lowered to a smaller of 4.5 V. The applied potential at the cathode vs the Ag/AgCl reference and total-cell voltage for the various applied current densities are listed in Table R1 and Table R2 respectively (reproduced below). The corresponding FE values are shown in Fig. R2.

We note that we have already previously also used this similar strategy of replacing the K₂SO₄ with a higher concentration of Cs₂SO₄ for our studies with simulated flue gas (Fig. 3), with similar outcomes of lowering total cell-voltage and increasing the C₂₊ FE.

Table R1 | Potential vs Ag/AgCl for CO₂R in 0.05 M H₂SO₄+1.5 M Cs₂SO₄ for Cu PTFE/Ni-N₄ at various applied cathodic current densities with pure CO₂ feedstock.

	Cu PTFE/Ni-N ₄
100 mA cm ⁻²	- 1.7 V
200 mA cm ⁻²	-2.0 V
400 mA cm ⁻²	-2.3 V
600 mA cm ⁻²	-3.0 V

Table R2 | Full-cell voltage for CO₂R in 0.05 M H₂SO₄+1.5 M Cs₂SO₄ of Cu PTFE/Ni-N₄ at various applied cathodic current densities with pure CO₂ feedstock.

	Cu PTFE/Ni-N ₄
100 mA cm ⁻²	3.2 V
200 mA cm ⁻²	3.9 V
400 mA cm ⁻²	4.5 V
600 mA cm ⁻²	5.1 V

Figure R2 | FE results of Cu PTFE/Ni-N₄ in 0.05 M H₂SO₄ + 1.5 M Cs₂SO₄ at various applied cathodic current densities with pure CO₂ feedstock.

2. The in-situ Raman studies are very informative to obtain insights into the reaction mechanism. CO(atop) is more active than CO(hollow+bridge) for C₂⁺ formation. In Fig 2d, the highest portion of CO_{atop} was found at -2.05V, while the faradaic efficiency of C₂⁺ peaked at -1.72 V. Did the authors consider other reasons besides CO adsorption for the selectivity of

C2+ formation? In addition, what is the influence on the surface CO population when oxygen is introduced, as is the case for flue gas?

Response

In Fig. 2d, the highest proportion of CO_{atop} was identified to be at -2.05 V vs Ag/AgCl. On the other hand, the optimal voltage for peak C₂₊ FE was identified to be at -1.72 V vs Ag/AgCl. Based on previous work by Li et al. (*Nature*, 2020, 577, 509–513), the authors observed a volcano correlation between the CO_{atop} to CO_{bridge} ratio to the C₂₊ FE. Similar in our case, we postulate that a similar situation could be occurring, with an optimal CO_{atop} to CO_{bridge} ratio occurring at -1.72 V vs Ag/AgCl, resulting in peak C₂₊ FE at this potential.

Besides CO adsorption, we reasoned that the intrinsic HER activity of the catalyst could also be an important factor that affects the C₂₊ FE. This is because the HER is a competing parasitic reaction during CO₂R. As a comparison, the HER performance of Cu PTFE/Ni-N₄ and Cu PTFE was tested in 0.5 M K₂SO₄ + 0.05 M H₂SO₄. As shown in Fig. R3, Cu PTFE/Ni-N₄ shows lower HER activity as compared to Cu PTFE. Specifically, Cu PTFE/Ni-N₄ has a higher overpotential of 362 mV to reach a current density of 10 mA cm⁻², as compared to only 292 mV with Cu PTFE. This is consistent with our CO₂R FE results, where Cu PTFE/Ni-N₄ exhibited a lower HER FE as compared to Cu PTFE.

Figure R3 | LSV curve of Cu PTFE/Ni-N₄ and Cu PTFE in 0.5 M K₂SO₄ + 0.05 M H₂SO₄.

An *in-situ* Raman spectroscopy study of Cu PTFE/Ni-N₄ was also performed in 0.05 M H₂SO₄ + 1.5 M Cs₂SO₄ electrolyte with simulated flue gas. This aimed to provide insights into the CO₂R mechanism under the influence of introduced O₂. As shown in Fig. S42a, the peaks at 970, and 1040 cm⁻¹ represent the O–O stretching vibration of O₂ (*Nat. Commun.* 2016, 7, 12440) and OH species (*J. Am. Chem. Soc.*, 2020, 142(2), 715–719) adsorbed on Cu PTFE/Ni-N₄. The existence of *O₂ and *OH species unambiguously confirms the occurrence of ORR on the catalyst surface. As for the *CO population, we only observed the presence of CO_{atop} with simulated flue gas (Fig. S42b). This situation appears to differ from the behaviour observed in pure CO₂ gas, where CO_{bridge} was also observed. We speculate that the absence of CO_{bridge} could be due to the competitive adsorption of ORR intermediates on the catalyst surface.

Figure S42 | (a) Potential resolved *in-situ* Raman spectroscopy of Cu PTFE/Ni-N₄ in 0.05 M H₂SO₄ + 1.5 M Cs₂SO₄ with simulated flue gas. (b) Raman heatmap of Cu PTFE/Ni-N₄ in the 2000–2120 cm⁻¹ region, showing the dynamic behavior of *CO.

3. The Cu PTFE/Ni-N₄ electrode is fabricated through a layer-by-layer strategy. Will the active site of Ni be covered by Cu, especially when the concentration of Ni is very low? Is there any systematic design strategy to ensure the exposure of the Ni active sites to proceed for tandem catalysis as suggested? In Fig 3.a, the Cu-PTFE/Ni-N₄ showed a higher ORR activity than Cu-PTFE alone. What is the rationale for choosing Ni-N₄? Did the authors consider choosing other catalysts with poorer ORR activity?

Response

For our catalyst, we sputtered 5 nm of Cu to construct each layer, which would most likely cover some of the Ni-N₄ sites. However, in our design principle, we reasoned that if this layer was too thin, there would be an inadequate amount of Cu active sites to further convert to CO generated by the Ni-N₄ catalyst. On the other hand, if this Cu layer was too thick then too many of the Ni-N₄ active sites would be covered and CO₂ to CO conversion would become compromised, resulting in a lower C₂₊ FE. Hence in our optimization process, we had prepared

electrodes with varying thicknesses of sputtered Cu layers (2, 5 and 10 nm). As illustrated in Fig. S20, we find that the total C₂₊ FEs are 47.0%, 75.2% and 55.9% for 2, 5 and 10 nm Cu respectively. Hence, through this empirical process, we found that the optimal Cu thickness was 5 nm, and this is why this value was chosen for our fabrication strategy.

Figure S20 | CO₂R performance in acidic electrolyte. The FE results for Cu PTFE/Ni-N₄ with different thickness (2, 5 and 10 nm) of copper layer. The black squares represent the total C₂₊ FE for each catalyst condition.

We found that Cu PTFE/Ni-N₄ does indeed give a higher ORR activity compared to Cu PTFE in alkaline media (Fig. 3a). However more importantly, this trend is reversed in acidic media, where Cu PTFE/Ni-N₄ has a lower ORR activity compared to Cu PTFE. While we do not know the exact reasons for this, we note that the acidic condition is more relevant to our work, for enabling production of C₂₊ products from simulated flue gas and suppressing ORR activity.

To perform CO₂-to-CO conversion, the choice of available known single-atom catalysts are Fe-N₄, Co-N₄ and Ni-N₄. Based on the literature, the Fe-N₄ is a catalyst that is very well-known to be active for ORR, hence we ruled out this catalyst (*Nanoscale* 2022, 14, 3212-3223; *J. Am. Chem. Soc.* 2019, 141, 36, 14115–14119; *Angew.Chem.Int. Ed.* 2021, 60, 25404–25410.). As for Co-N₄, this is a catalyst system that is often investigated for good HER activity (*Adv. Funct. Mater.* 2021, 31, 2100547; *Adv. Energy Mater.* 2019, 9, 1803689; *Angew. Chem.Int. Ed.* 2022, 61, e20211495.). Since HER is a competitive side reaction to CO₂ reduction, we also opted not to use this catalyst. Finally, Ni-N₄ is a widely reported catalyst that is known to exhibit a very high CO Faradaic efficiency (FE) near to 100 % without HER (*Adv. Mater.* 2022, 34, 2201295; *Angew. Chem. Int. Ed.* 2021, 60, 7607; *Angew. Chem. Int. Ed.* 2020, 59, 20589.). Furthermore, the ORR activity of Ni-N₄ is also known to be poor (*Nat. Commun.* 12, 5589 (2021); *Angew. Chem. Int. Ed.* 2021, 60, 4448; *ACS Catal.* 2014, 4, 10, 3797–3805.), hence we selected Ni-N₄ as the catalyst of choice to construct the tandem electrocatalyst system reported in this work.

Minor comments:

Typos in Line 222. I think the authors meant Fig 2d rather than Fig 2f.

Response

The corresponding typos have been corrected in the revised manuscript.

Once again, we sincerely thank Reviewer #2 for their helpful comments.

Reviewer #3 (Remarks to the Author):

In this manuscript, the authors describe a strategy for the generation of multicarbon products from simulated flue gas, where acidic electrolyte was used to significantly suppress ORR on a Cu PTFE/Ni-N₄ electrode. The authors achieve FEs toward multicarbon products of 82.4 % and 46.5% in pure CO₂ gas and in simulated flue gas respectively. This was supported theoretically using DFT simulations where they found increases in the free energy change of the rate-determining steps for ORR on Cu and Ni-N₄ in acidic media. This manuscript is suitable for publication in Nature Communications after appropriate revision, with detailed comments as given below:

Response

We thank Reviewer #3 for taking the time to evaluate our manuscript and providing helpful suggestions to improve our manuscript.

1. More statistical analysis of the Ni-N₄ size should be provided, which in the ideal case should yield a distribution close to a Gaussian shape.

Response

We have conducted a re-evaluation of the particle size distribution. The updated size distribution outcomes are presented as below:

Figure 1c | TEM images of Ni-N₄, consisting of Ni single-atoms hosted on a carbon support.

2. More information should be provided regarding the computational methods, such as why the top two layers of Cu (111) were chosen to move freely, the criteria for selecting K-points, and whether spin polarization was considered.

Response

The reason why the top two layers of Cu (111) were chosen to move freely is because the interactions between the adsorbates and the two lower layers are negligible compared to that between the adsorbates and the top two layers. Fixing the two lower layers can reduce computational demand and expedite the computation process. Therefore, we used a 4-layer Cu (111) model with the two upper layers relaxed and the two lower layers fixed.

Choosing an appropriate K-point grid density is crucial. Higher density yields more accurate results but at a higher computational cost. In this work, we start with a lower K-point density and gradually increase it until the results converge, with no significant changes. To ensure convergence of the k-point grid size, we evaluated the overall energy of Cu (111) from 1 1 1 to the 8 8 1 k-point grid. The grid size was increased to 4 4 1, achieving convergence. Therefore, in the Cu (111) system, the k-point grid is 4 4 1 (Fig. R4).

For the spin polarization, we set ISPIN = 1 and ISPIN =2 in Cu (111) and Ni-N₄ system, respectively.

In addition, the missing computational parameters have been added in the revised manuscript.

Figure R4 | Accurate geometry relaxation of the Cu (111) slab model.

3. Did the authors test the performance of Cu PTFE in 1 M Cs⁺ and 3 M Cs⁺ solutions for CO₂R with flue gas?

Response

The CO₂R performance of Cu PTFE in 1 M Cs⁺ and 3 M Cs⁺ solutions with flue gas have been tested (Fig. S39-40) and added into the Supplementary Information as Fig. S39 and Fig. S40 respectively. Similarly, when 1 M K⁺ is changed to 1 M Cs⁺, the C₂₊ product FE increased from 5.8% to 10.8% respectively. Increasing the concentration to 3 M Cs⁺ resulted in a further increase in the C₂₊ product FE to 15.7%. As mentioned previously in the manuscript, this is because a higher Cs⁺ ion concentration is better at promoting C-C coupling (*J. Am. Chem. Soc.* 2017, 139, 32, 11277–11287; *J. Am. Chem. Soc.* 2016, 138, 39, 13006–13012; *J. Am. Chem. Soc.* 2017, 139, 45, 16412–16419; *ACS Catal.* 2018, 8, 11, 10012–10020).

Figure S39 | FE results of Cu PTFE in 0.05 M H₂SO₄ + 0.5 M Cs₂SO₄ at different current densities with simulated flue gas.

Figure S40 | FE results of Cu PTFE in 0.05 M H₂SO₄ + 1.5 M Cs₂SO₄ at different current densities with simulated flue gas.

4. In Figure S20, the authors claim that the CO₂R performance of Cu PTFE/Ni-N₄ surpasses that of Cu PTFE/Ni-N₄-1 and Cu PTFE/Ni-N₄-2. What will happen to the performance if more layers are added to the catalyst system?

Response

In our optimization process, we empirically found 3 layers to give the optimal performance for C₂₊ product FE. In Figure S21 we show the CO₂R performance obtained when an additional 4th layer of Ni-N₄ + sputtered Cu layer is added (Cu PTFE/Ni-N₄-4). We found that with 4

layers, the total C₂₊ FE decreases to a value 61.3%. This is the reason why we selected 3 layers, which gave the best performance results (Fig. S21). We reason that this is because too many layers could result in CO₂ mass transport limitations.

Figure S21 | CO₂R FE results in 0.5 M K₂SO₄ + 0.05 M H₂SO₄ for Cu PTFE/Ni-N₄-1, Cu PTFE/Ni-N₄-2, Cu PTFE/Ni-N₄, and Cu PTFE/Ni-N₄-4 at 200 mA cm⁻².

5. Please elucidate the procedure employed by the authors to determine the area/proportion of *CO_{hollow}, *CO_{bridge}, and *CO_{atop}, as well as how they validated the ratio of *CO_(hollow + bridge) to *CO_{atop}? Is this method an original development, or has it been previously established? If this is not a new method, kindly provide additional details and cite the relevant research articles.

Response

The analysis method for the *in-situ* Raman spectroscopy results in this paper is based on prior literature (*Nature* 577, 509–513 (2020); *Nat. Nanotechnol.* 18, 299–306 (2023); and *Nat Catal* 6, 319–331 (2023)). Specifically, each spectral band was deconvoluted into individual bands for *CO_{atop}, *CO_{hollow} and *CO_{bridge} using Lorentzian fitting. We have added more details of the analysis method in the manuscript.

6. Authors should check through the manuscript for mistakes, for example there are some errors in subscripts/superscripts.

Response

The corresponding errors have been corrected in the revised manuscript.

Once again, we would like to thank the reviewer for providing helpful suggestions to improve our work.

Reviewer #4 (Remarks to the Author):

The manuscript entitled “Acidic media enables Oxygen-tolerant Electrosynthesis of Multicarbon Products from Simulated Flue Gas” describes the discovery that the choice of electrolyte and catalyst design can be used to enable oxygen-tolerant production of C₂+ products in simulated flue gas. The work presented in the manuscript has some novelties and the idea is interesting and relevant to the needs of the current CO₂ reduction experiments. The manuscript is well organized, however, there are still some issues that need to be addressed.

Response

We are thankful to Reviewer #4 for their positive evaluation of our manuscript.

Subject comments

1. In the abstract and introduction section, abbreviations Ni-N₄ need to be explained.

Response

We named the Ni-based single-atom catalyst according to their active site configuration, which is Ni-N₄. We have replaced Ni-N₄ with the term “carbon supported single-atom Ni” in the abstract and introduction section to avoid confusion. We later term this as Ni-N₄ in the results and discussion section.

2. Although the author claims the catholyte is acidic (0.05M H₂ SO₄ + 0.5 M K₂SO₄ or 1.5 M Cs₂SO₄), it would be nice to know about the operating pH conditions throughout the experiment.

Response

During CO₂ reduction, protons are either consumed or hydroxide anions generated. Hence, the local pH at the electrode surface will become higher than that of the bulk under CO₂ reduction conditions (*J. Electroanal. Chem.*, 2006, 594, 1–19). To explore this, there has been a report by Lu et al. (*J. Am. Chem. Soc.*, 2020, 142, 15438–15444), which demonstrates that *in-situ* Raman spectroscopy can be employed to experimentally deduce the local pH under reaction conditions. Specifically, the authors used Raman spectra to assess the relative concentrations of HCO₃⁻ and CO₃²⁻ at the electrode surface. Subsequently, local pH values were determined by assuming steady-state concentrations and using the known acid-base equilibrium constants of HCO₃⁻ and CO₃²⁻.

Here, we also conducted *in-situ* Raman spectroscopy tests on the catalyst with the 0.05 M H₂SO₄ + 0.5 M K₂SO₄ electrolyte which has a bulk pH of 1.7. In the voltage range of -1.09 to -2.05 V vs Ag/AgCl, we did not observe the presence of HCO₃⁻ or CO₃²⁻ peaks in the Raman spectrum (Fig. S30a). Since these peaks are absent, we are unable to derive an exact local pH value. However, this absence suggests that the local pH is likely acidic, since neutral/alkaline conditions are required for HCO₃⁻ or CO₃²⁻ to be present.

To study this further, we repeated the same experiments with the 0.05 M H₂SO₄ + 0.5 M K₂SO₄ electrolyte, except 1 M KOH was added to this until the bulk pH reached a value of 2.2. With a bulk pH of 2.2, peaks in the Raman spectrum related to HCO₃⁻ and CO₃²⁻ continued to be absent at the open-circuit potential (OCP) and -1.09 V vs Ag/AgCl. This suggests that the local pH remains acidic under these conditions. However, beginning at -1.26 V vs Ag/AgCl, we started to observe the HCO₃⁻ and CO₃²⁻ peaks in the Raman spectra (Fig. S30b). These peaks

are also present at the more negative potentials. This suggests that the local pH is no longer acidic, due to consumption of protons and generation of hydroxide anions.

It is well known that in an acidic solution, the critical pH for the hydrolysis of carbon dioxide (CO₂) depends primarily on the acid-base equilibrium in water, particularly the equilibrium reaction between CO₂ and water. CO₂ can undergo hydrolysis with water to form carbonic acid (H₂CO₃), which can then further dissociate into bicarbonate ions (HCO₃⁻) and protons (H⁺):

In this reaction, the critical pH in an acidic solution typically refers to the pH at which the concentrations of H⁺ ions and HCO₃⁻ ions are equal. This corresponds to the balance between carbonic acid and its dissociation products. At room temperature, this critical pH value is usually around 6.35. Additionally, CO₂ typically does not undergo hydrolysis into carbonic acid when the pH is less than approximately 4.5 (*J. Geosci. Educ.*, 2002, 50, 357-362). This is because at lower pH conditions, the increased concentration of hydrogen ions in the water promotes the equilibrium reaction between bicarbonate ions and carbon dioxide to shift to the left, resulting in more CO₂ formation rather than the formation of carbonic acid.

Based on this analysis and our *in-situ* Raman spectroscopy results, we thus concluded that the local pH of our electrode under CO₂R conditions should be below 4.5 and hence remains acidic when an electrolyte of 0.05M H₂SO₄ + 0.5 M K₂SO₄ (bulk pH 1.7) is used. On the other hand, when the bulk pH was adjusted to 2.2, the local pH can no longer be maintained in an acidic region under CO₂R conditions.

Figure S30 | Raman spectra recorded at various pH value. (a) bulk: pH = 1.7 (b) bulk: pH = 2.2. HCO₃⁻ and CO₃²⁻ peaks are recorded at 1012 and 1064 cm⁻¹. Voltages are reported vs Ag/AgCl. OCP stands for open-circuit potential, where no current is passing through the system.

3. The XPS spectrum in Fig S4, Fig S14 should be assigned. The survey spectrum shows some other peaks that should be identified. The high-resolution spectrum needs to be fitted properly to find out the oxidation states.

Response

According to this suggestion, we have fitted the high-resolution XPS spectrum and updated Fig. S4, S9, S14, S23 (all reproduced below). For Cu 2p, it is well-known that Cu⁺ and Cu⁰

have very similar positions, hence it is difficult to deconvolute their individual contributions. Hence, we do not provide the relative proportions of $\text{Cu}^0:\text{Cu}^+:\text{Cu}^{2+}$. For the Ni- N_4 catalyst, we have deconvoluted the relative contributions of pyridinic, graphitic, oxidized and pyrrolic N (Fig. S9b, Fig. S14b and Fig. S23b).

Figure S4 | (a) Narrow scan X-ray photoelectron spectroscopy (XPS) spectrums of Cu 2p and (b) survey scan of Cu PTFE.

Figure S9 | X-ray photoelectron spectroscopy (XPS) characterization of Ni- N_4 . Narrow scans of (a) Ni 2p, (b) N 1s and (c) C 1s. (d) survey scan.

Figure S14 | X-ray photoelectron spectroscopy (XPS) characterization of Cu PTFE/Ni-N₄. Narrow scans of: (a) Cu 2p, (b) N 1s and (c) Ni 2p. (d) survey scan.

Figure S23 | X-ray photoelectron spectroscopy (XPS) characterization of Cu PTFE/Ni-N₄ after CO₂R. Narrow scans of: (a) Cu 2p, (b) N 1s and (c) Ni 2p. (d) survey scan.

4. In Fig S6, and Fig 2 why the overall FE is not 100%, if it's not 100%, does the author expect any other products? Then in Figure S11, shows that the overall FE is over 100%. Why those discrepancies?

Response

In CO₂R experiments, the overall FE is a summation of all detected gas and liquid products. There is no definitive guideline in the field, however total reported FEs should ideally be at least >90% for reliable results. This is because minor products could be generated, which were not detected or analyzed. In addition, some liquid products could also be more volatile and hence losses could easily occur. Products such as ethanol could also migrate to the anode and oxidize, further contributing to losses. In addition, there could also be experimental and human errors which contribute to the overall FE not being a 100%. Hence typically, results are reported with error bars indicating the standard deviation from three independent measurements to ensure reliable results. In our work, our total FE generally lies close to a 100%. Fig. R5 are examples taken from the literature, which illustrates that such variations are common in CO₂R tests.

Figure R5 | Total FE results from the literature. (a) and (b) are from *Nat. Nanotechnol.*, 2023, 18, 299–306 and (c) is from *Nat. Catal.*, 2020, 3, 98–106.

5. The author produced the EIS spectrum in Figure S18, but there is no significant change in the Rct value of the three catalysts to claim that the Ni-N4 catalyst possesses faster kinetics. Thus, the claim is ambiguous.

Response

We have carefully considered this statement again and agree with the reviewer. To avoid any misunderstanding, we have removed the sentence related to "faster kinetics".

6. The authors performed DFT simulations on ORR activity, however in acidic conditions a DFT calculation about CO₂RR with hydrogen evolution path will give more insights.

Response

The reviewer has raised a good point regarding the HER activity, which tends to compete with CO₂R and hence is a parasitic side reaction. Although the ORR FE becomes suppressed in acidic electrolyte, we observed that the HER FE actually becomes slightly increased in acidic electrolyte as compared to alkaline electrolyte in our simulated flue gas experiments (Fig. 3c and Fig. S36). For example, at a current density of 200 mA cm⁻², the HER FE for Cu PTFE/Ni-N₄ has a value of 20.6% in alkaline electrolyte and 30.6% in acidic electrolyte. This is expected

because HER in acidic electrolyte directly consumes protons and hence avoids the water dissociation step which is known to be rate-limiting in alkaline HER (*Adv. Funct. Mater.*, 2021, 31, 2101578) (*Energy Environ. Sci.*, 2021, 14, 5228-5259) (*Adv. Energy Mater.*, 2019, 9, 1901333). As a result, performing CO₂R in acidic media without alkali metal cations is known to result in nearly 100% FE for HER (*Science*, 2021, 372, 1074–1078). With the addition of alkali metal cations, Huang et al. showed that the HER activity could be suppressed and hence reasonable FEs towards C₂₊ products could be achieved in acidic electrolyte through addition of a high concentration of alkali metal cations (*Science*, 2021, 372, 1074–1078), which we have also done in this work.

Following the reviewer's suggestion, we had initially considered to perform DFT calculations to understand this HER activity suppression effect by these alkali metal cations. However, upon reviewing the literature we found that explaining this effect cannot be adequately accomplished through DFT calculations. Instead, this effect might be explained better through Poisson–Nernst–Planck (PNP) simulations (*Nat. Catal.*, 2022, 5, 268–276). For instance, work by Gu et al. (*Nat. Catal.*, 2022, 5, 268–276) found that the reason for reduced HER activity in their acidic electrolyte is due to the high concentration of hydrated alkali cations that they employed (K⁺ cations). Using a Poisson–Nernst–Planck (PNP) simulation model, the authors found that hydrated alkali cations physisorbed on the cathode modify the distribution of electric field in the double layer, which then impedes HER by suppressing the migration of hydronium ions. Similarly, in our case the acidic electrolyte employed contains a high concentration of alkali metal cations (K⁺ and Cs⁺). Hence, we reasoned that a similar effect likely operates in our system to suppress the HER activity for operating in both pure CO₂ and simulated flue gas feedstock with acidic electrolyte.

7. What is the practical relevance of this single atom catalyst? How feasible is it to upscale it and bring to practical use beyond the lab scale study?

Response

The practical relevance of single atom catalysts (SACs) is the potential to achieve near 100% atomic utilization, meaning that every single metal atom in the catalyst could potentially serve as an active site (*Electrochem. Energ. Rev.*, 2019, 2, 539–573) (*Nat. Rev. Chem.*, 2018, 2, 65–81) (*Chem.* 2019, 5, 2733–2739). This is in contrast to bulk films or nanoparticles, where metallic atoms that are buried in the subsurface typically do not participate in facilitating the reaction of interest and are hence inactive. This high atomic utilization is especially important for noble metals such as Pt, which tend to be expensive and scarce. In addition to this, SACs tend to have very different catalytic properties from their bulk counterparts. For instance, bulk Ni is inactive for CO₂R, but Ni SAC can selectively convert CO₂ to CO.

Our catalyst synthesis method is facile, consisting of a hydrothermal procedure, followed by metal impregnation and thermal annealing and could potentially be amenable to scale up. Although we did not attempt to scale up our synthesis process, we turn to the literature where studies on SAC synthesis scale up have been performed. For instance, Yang et al. (*Nat. Commun.*, 2019, 10, 4585) introduced a universal and robust ligand-mediated approach for synthesizing M-SACs with high metal content. This method is versatile and applicable to the synthesis of M-SACs containing first-, second-, and third-row transition metals on carbon supports. Furthermore, it leverages commercially available conductive carbons as a support and facilitates large-scale SAC production, reaching the kilogram scale. Additionally, Sun et al. (*Nat. Commun.*, 2023, 14, 1599) reported a microwave-assisted strategy for fabricating coordinatively unsaturated metal-nitrogen sites doped within defective carbon nanotubes (Fe,

Co, or Ni-CNTs-MW). This simple synthetic approach is universal for gram-scale production within approximately 2 minutes. These studies therefore indicate the potential for further scale-up of SACs for practical deployment.

More relevant to CO₂R, Zheng et al. (*Joule*, 2019, 3, 265–278) synthesized single-atom Ni catalysts for CO₂ to CO conversion and deployed this for use in a large area 100 cm² membrane electrode assembly CO₂ electrolyzer system. They demonstrated long-term electrolysis under a full-cell voltage of 2.8 V and a current of 8 A. At the same time, the CO selectivity maintained above 90% over the course of 6 hours of continuous operation. Hence, this study indicates that these SACs could potentially be synthesized and deployed on a larger scale. Based on these studies, we believe that there is a promising route to bring SACs to practical use beyond lab scale studies.

8. What about testing with real flue gas which will give a more realistic case?

Response

Real flue gas consists of non-negligible amounts of impurities such as SO_x and NO_x. For the case of SO_x, this is known to contaminate the Cu surface (*J. Am. Chem. Soc.*, 2019, 141, 25, 9902–9909) and cause an irreversible switch of the reaction pathway entirely to formate production. As for NO_x, previous work (*Nat. Commun.*, 2020, 11, 5856) has shown that although its effect is reversible, NO_x reduction can compete with CO₂ reduction and hence can consume a good portion of the current. In this work, the focus was to overcome the dual challenges of low CO₂ concentration and the competing ORR. Hence, we employed simulated flue gas to study this in a systematic and reproducible way. In future work, we are currently indeed actively researching on ways to address the SO_x/NO_x issue. Once this is resolved, directly employing real flue gas would become possible. In addition, we are currently in active talks with industry partners who can supply us with real flue gas.

Once again, we would like to sincerely thank the reviewer for their constructive comments which have helped us improve the quality of our work.

REVIEWER COMMENTS

Reviewer #2 (Remarks to the Author):

All my concerns are properly addressed.

Reviewer #3 (Remarks to the Author):

I can see the authors have conducted additional research and discussions to address the concerns properly. Thus I would recommend acceptance at this stage.

Reviewer #4 (Remarks to the Author):

In the revised version, the authors seem to respond appropriately to what we have asked. The manuscript, SI, and experimental data were addressed point-by-point. The defects in the main text, SI, figure captions, and figures were modified.

However, the discussion about XPS seems to be weak and the fitted curves in Figure S14 and S23 is not appropriate.

We are still concerned about the DFT calculation. The authors keep the assumption that the missing FE can be assigned to ORR which cannot be true as there will be a hydrogen pathway. However, we are also concerned about the originality of this work because there are previous reports on similar work. The authors need to clarify this point to emphasize this work's idea and strong point.

Except for this concern, the authors have answered and revised many things to improve the quality of the paper. We believe this paper might be accepted for publication if the interpretations related to DFT calculation are addressed and the originality is clarified.

Once again, we thank the editor for handling our manuscript and the reviewers for their comments. Changes within the manuscript and supporting information are highlighted in yellow. Below is a response to the reviewer comments, which are written in blue font.

Reviewer #2 (Remarks to the Author): All my concerns are properly addressed.

Response

We are thankful to Reviewer #2 for their constructive advice and suggestions.

Reviewer #3 (Remarks to the Author): I can see the authors have conducted additional research and discussions to address the concerns properly. Thus I would recommend acceptance at this stage.

Response

We are grateful to Reviewer #3 for their time and helpful comments.

Reviewer #4 (Remarks to the Author):

In the revised version, the authors seem to respond appropriately to what we have asked. The manuscript, SI, and experimental data were addressed point-by-point. The defects in the main text, SI, figure captions, and figures were modified.

Response

We would like to thank Reviewer #4 for taking the time to evaluate our revised manuscript.

1. However, the discussion about XPS seems to be weak and the fitted curves in Figure S14 and S23 is not appropriate.

Response

We have refitted and updated the XPS data in Fig. S14 and S23 (reproduced below). Specifically, the Cu 2p_{3/2} XPS fitting curves revealed that Cu was mainly in the oxidized state in Cu PTFE/Ni-N₄. Our Raman spectroscopy results (Fig. S5) also corroborates the presence of oxidized Cu on the sample surface. Since the Cu films are prepared by sputtering, they should be in the metallic form in the pristine as prepared condition. These oxidized forms of Cu are a result of oxidation of the exposed Cu surface in ambient air, which reduce to its metallic form upon application of a cathodic potential during CO₂ reduction (*Angew. Chem. Int. Ed.* 2018, 57, 551-554) (*Appl. Surf. Sci.* 2018, 255, 2730–2734) (*ACS Appl. Mater. Interfaces* 2018, 10, 10, 8574–8584). Discussion of the XPS results has now been included in the manuscript.

We note that the XPS data are included as part of the catalyst characterization process. However, due to oxidation of the catalyst in ambient air, there is no clear relevance between this XPS data and explaining the catalyst activity/performance. For this reason, we did not provide any in-depth discussion of the XPS results in the previous manuscript versions.

Figure S14 | X-ray photoelectron spectroscopy (XPS) characterization of Cu PTFE/Ni-N₄. Narrow scans of: (a) Cu 2p, (b) N 1s and (c) Ni 2p. (d) survey scan.

Figure S23 | X-ray photoelectron spectroscopy (XPS) characterization of Cu PTFE/Ni-N₄ after CO₂R. Narrow scans of: (a) Cu 2p, (b) N 1s and (c) Ni 2p. (d) survey scan.

2. We are still concerned about the DFT calculation. The authors keep the assumption that the missing FE can be assigned to ORR which cannot be true as there will be a hydrogen pathway.

Response

We clarify that we actually do indeed quantify the FE towards the hydrogen pathway (hydrogen evolution reaction HER) and report these results together with the CO₂ reduction products. For instance, in Fig. 3c (reproduced below) the HER FE is shown in light grey and the ORR FE is shown in dark grey. Exact FE values and full results are also provided in Tables S14-S40.

Figure 3c | Product FE for Cu PTFE in 1 M KOH and Cu PTFE/Ni-N₄ in 0.05 M H₂SO₄ + 1.5 M Cs₂SO₄ under different current densities.

In our CO₂R experiments with flue gas, both HER and ORR serve as competing reactions. We are able to measure the HER FE using gas chromatography. However, it is challenging to determine the ORR FE because we operate in aqueous electrolyte and the product generated is water. Therefore, we have to assume that the missing FE must be attributed to ORR. We note that this assumption has also been employed in recent publications (*J. Am. Chem. Soc.* 2023, 145(48), 25933–25937) (*Energy Environ. Sci.* 2020, 13, 554-561).

According to this suggestion, we have also performed additional DFT calculations of the hydrogen pathway. It is known that the dissociation of water is the rate-determining step for HER in alkaline media (*Nat. Commun.* 2021, 12, 6766) (*Adv. Mater.* 2022, 34, 2108505) (*Adv. Sci.* 2021, 8, 2001881) (*Adv. Funct. Mater.* 2021, 31, 2103673). As for acidic media, the Gibbs free energy of H* (ΔG_{H^*}) is usually considered as an effective standard to evaluate HER activity (*Adv. Energy Mater.* 2020, 10, 2002260) (*J. Mater. Chem. A* 2021, 9, 10326-10334) (*Nat. Commun.* 2022, 13, 1189). Based on our DFT calculations, we found that the Ni-N₄ system exhibits energy barriers of 2.56 eV in alkaline media and 1.57 eV in acidic media for HER (Fig. S32-S33). As for Cu (111), energy barriers of 0.23 eV in alkaline media and 0.12 eV in acidic media (Fig. S32-33) were determined. These results suggest that HER is easier in acidic media as compared to alkaline media and is consistent with expectations (*Adv. Energy Mater.* 2020, 10, 2002260) (*Adv. Mater.* 2021, 33, 2007894). This is also consistent with our FE data where the HER FE for Cu PTFE/Ni-N₄ was found to be 20.6% in alkaline electrolyte (Fig. S37) and 30.6% in acidic electrolyte (Fig. S39) at a current density of 200 mA cm⁻². However, we note that even though HER activity is significantly higher in acidic media, the presence of a high concentration of alkali metal cations (e.g. K⁺) can help to suppress the HER activity and promote CO₂ reduction. This was previously demonstrated by Huang et al. (*Science*, 2021, 372, 1074–1078) and also successfully employed in our work.

Figure S32 | Alkaline HER free energy diagram of (a) Ni-N₄ and (b) Cu (111). The red dotted line represents the activation energy barrier for the rate-determining step.

Figure S33 | Reaction pathway for proton reduction to hydrogen on Ni-N₄ and Cu (111), involving *H as the intermediate. The red dotted line represents the free energy change for the rate-determining step.

3. However, we are also concerned about the originality of this work because there are previous reports on similar work. The authors need to clarify this point to emphasize this work's idea and strong point.

Response

We would like to emphasize that the novelty of the work is on identifying an electrolyte composition and electrocatalyst system combination that can enable the generation of multicarbon (C₂₊) products directly from simulated flue gas with reasonable FEs. In this process, we needed to solve the two main challenges involved, which is the low CO₂ concentration (15%) and suppressing the competing oxygen reduction reaction. For the first time, we have successfully demonstrated in this work that acidic electrolyte in combination with a Cu PTFE/Ni-N₄ tandem electrocatalyst system can solve these challenges.

To the best of our knowledge, there is only work by Xu et al. (*Energy Environ. Sci.* 2020, 13, 554-561) that investigates the direct utilization of simulated flue gas (N₂/CO₂/O₂ mixture) for generation of multicarbon products. In this work, they employed a selective CO₂ mass transport strategy by using ionomer coatings on their Cu catalysts which selectively slowed down O₂ mass transport. However, we note that this required high pressure conditions of 10 bar to obtain reasonable FEs to C₂₊ products. In our work, high pressure conditions are not required and ambient pressure was sufficient. Furthermore, our simple yet effective strategy involving electrolyte selection and electrocatalyst design is completely different and novel.

Below is a comparison of our work with previous literature involving electrochemical CO₂ reduction with mixtures of CO₂/O₂.

Work by He et al. (*Nat. Commun.* 2020, 11, 3844) found that the partial current density to C₂₊ products could be increased in CO₂/O₂ mixtures as compared to pure CO₂ alone, especially at the lower overpotentials. However, it is important to note that as a result of the O₂ addition,

even though the C_{2+} partial current density increases, the total FE actually decreases significantly. This is because a significant amount of current becomes consumed by ORR with addition of O_2 , such that the FE towards CO_2R products becomes very small. This observation is also consistent with our result in alkaline media (Fig. 3c), where we found that a significant portion of the current is consumed by ORR and very little FE towards CO_2R products. Importantly, we show in our work that acidic electrolyte can suppress the ORR and allow for reasonable FE values towards C_{2+} products.

Work by Lu et al. (*Sci. Bull.* 2019, 64, 1890-1895) and Li et al., (*Angew. Chem. Int. Ed.* 2020, 59, 1091810923) utilized a selective CO_2 mass transport strategy by coating their cobalt phthalocyanine or Sn catalysts with a selectively permeable polymer with a high CO_2/O_2 selectivity. As a result, they achieved a 75.9% Faradaic efficiency (FE) for CO production when a feedstock containing 5% O_2 and 95% CO_2 was employed. For the Sn catalyst, they generated formate with nearly 100 % selectivity and a current density of 56.7 mA cm^{-2} with a feedstock of 95% CO_2 and 5% O_2 . Importantly however, they did not demonstrate if their strategy could be applied to the generation of C_{2+} products on Cu. Furthermore, in our case the CO_2 concentration we used was much lower at only 15%. We also note that it very important to develop other strategies in addition to selective CO_2 mass transport that could be simple yet effective, which we successfully demonstrated our strategy involving electrolyte selection and electrocatalyst design. This is because in future work, our strategy could be potentially combined with the selectively permeable polymer strategy to yield even better results.

These works are cited and discussed in our manuscript.

Once again, we are grateful to the reviewer for these helpful suggestions to improve our work.

REVIEWERS' COMMENTS

Reviewer #4 (Remarks to the Author):

Pleased to read this revised version of the manuscript and notice that the authors have carried out the revisions suggested to them. Indeed, the revised manuscript looks much better than the original submission. Also, the authors have answered very clearly all the concerns raised by me as and other reviewers as well. For my comments, I have no further remarks to make on this manuscript and I am satisfied with the current revised version.